# Using Machine Learning Algorithms Based on GF-6 and Google Earth Engine to Predict and Map the Spatial Distribution of Soil Organic Matter Content

Zhishan Ye [1], Ziheng Sheng [1], Xiaoyan Liu [1], Youhua Ma [1], Ruochen Wang [2], Shiwei Ding [1], Mengqian Liu [3], Zijie Li [4] and Qiang Wang [1],*

1. College of Resources and Environment, Anhui Agricultural University, Hefei 230036, China; yezhishan@stu.ahau.edu.cn (Z.Y.); shengzh@stu.ahau.edu.cn (Z.S.); liuxy@stu.ahau.edu.cn (X.L.); yhma@ahau.edu.cn (Y.M.); 18720454@stu.ahau.edu.cn (S.D.)
2. Heinz College of Information Systems and Public Policy, Carnegie Mellon University, Pittsburgh, PA 15213, USA; ruochenw@andrew.cmu.edu
3. School of Plant Protection, Anhui Agricultural University, Hefei 230036, China; liumq@stu.ahau.edu.cn
4. Realty Research Center, Nanjing Agricultural University, Nanjing 210095, China; zjli@stu.njau.edu.cn
* Correspondence: 28104@ahau.edu.cn

**Abstract:** The prediction of soil organic matter is important for measuring the soil's environmental quality and the degree of degradation. In this study, we combined China's GF-6 remote sensing data with the organic matter content data obtained from soil sampling points in the study area to predict soil organic matter content. To these data, we applied the random forest (RF), light gradient boosting machine (LightGBM), gradient boosting tree (GBDT), and extreme boosting machine (XGBoost) learning models. We used the coefficient of determination ($R^2$), root mean square error (RMSE), and mean absolute error (MAE) to evaluate the prediction model. The results showed that XGBoost ($R^2 = 0.634$), LightGBM ($R^2 = 0.627$), and GBDT ($R^2 = 0.591$) had better accuracy and faster computing time than that of RF ($R^2 = 0.551$) during training. The regression model established by the XGBoost algorithm on the feature-optimized anthrosols dataset had the best accuracy, with an $R^2$ of 0.771. The inversion of soil organic matter content based on GF-6 data combined with the XGBoost model has good application potential.

**Keywords:** geospatial modeling; machine learning; predictive mapping; remote sensing inversion; soil organic matter

## 1. Introduction

Soil organic matter (SOM) is an important factor considered in soil surveys and environmental quality assessments [1–4], a key factor that participates in the global carbon cycle [5–7], and is an important indicator for judging the level of soil fertility [8,9]. Determining the SOM content is vital for achieving sustainable agricultural development [10] and ecological civilization [11], supporting ecosystem services [12] and improving crop productivity [13,14].

With the increasing demand for informatization for precision agriculture, accurately, quickly, and extensively estimating the SOM content has become challenging for many researchers. The traditional geostatistical method for predicting SOM content involves the measurement of SOM content in many samples, with the help of scale deductions and related geostatistical models [15–17]; the core theories are variogram and kriging interpolation [18]. Although the geostatistical method is reliable in theory, its main limitation in predicting the SOM content is that the sampling dataset has difficulties in meeting the stationarity assumption under complex terrain environments, and a large amount of data is needed to meet the needs of spatial autocorrelation. Collecting many soil samples in a research area inevitably leads to problems such as a long sampling period, high

economic cost, and low prediction accuracy of a single data source [19,20]. With the development of high-resolution satellite remote sensing technology, researchers have found that different SOM contents have unique spectral response characteristics in the visible and infrared bands [21]. Researchers combine spectral characteristics with SOM content using remote sensing technology. Retrieving SOM content is, thus, a hot topic in soil science research [22–28].

The quantitative inversion of the reflectance spectra of SOM began in the 1960s [29]. Researchers have found a correlation between the spectral reflectance of the soil and SOM content. The laboratory-based prediction of SOM content can only provide point-scale prediction data [30–33], not landscape-scale prediction data. Satellite remote sensing can be used to produce real-time, dynamic, macroscopic, accurate, and low-cost predictions. The dynamic monitoring of SOM has received extensive attention from the soil science community [22–25]. One of the best options for the inversion of SOM content is considered to be establishing a quantitative SOM inversion model using ground-measured data and multi-spectral satellite remote sensing data with a high spatial resolution, spectral resolution, rich information, and high positioning accuracy to predict large-scale SOM content [26,27]. The SOM content of soil presented in a remote sensing image can be predicted by a regression model. This Model established a relationship between the spectral reflectance and SOM content of ground samples. When putting the spectral reflectance information of non-sampling points into it, the SOM content is the result, which was obtained from the calculation in the regression equation.

Since the 1970s, with the use of Landsat data, multispectral satellite data have been widely used in soil surveys [34–37]. Researchers have begun to combine satellite remote sensing data and soil data to establish a regression relationship between soil band reflectivity and organic matter content to predict large-scale SOM content. Different regression algorithms and mathematical transformations of reflectivity [38–40] affect the accuracy of organic matter prediction. The SOM inversion method is not universal for all regions and performs differently in different practical applications. Therefore, remote-sensing inversion of SOM is another research hotspot [41–43]. Machine learning algorithms have been gradually introduced for the prediction of various soil properties in the fields of mathematics and computers [44–47]. Numerous experiments have proven that machine learning algorithms perform well in analyzing nonlinear SOM characteristics and are more effective for multi-source and multi-feature data [22,46–49]. Compared with traditional linear regression algorithms, machine learning algorithms have a higher prediction accuracy and faster running speed for SOM inversion. Support vector machines and random forest (RF) algorithms have been widely used in previous single-element SOM content inversions [22,46–50]. With the update and iteration of the machine learning framework, models with higher prediction accuracy and better performance, such as the light gradient boosting machine (LightGBM) [45,51], gradient boosting tree (GBDT) [52–54], and extreme gradient boosting machine (XGBoost) [44,55,56] are now being widely used in agriculture, although rarely for SOM content prediction and research.

In this study, we selected Hefei City, Anhui Province, China as the research area to explore the prediction accuracy of SOM based on multi-source data, such as multi-spectral data, elevation, slope, vegetation index, cultivated land planting situation, and soil type. Our aim was to find the most accurate, fastest, and most stable SOM content inversion method suitable for farmland soil in this research area, to provide a valuable reference for the spatial estimation of SOM content.

## 2. Materials and Methods

### 2.1. Overview of the Research Area

The study area is located in Hefei, the capital city of Anhui Province, in eastern China, and is the sub-central city of the Yangtze River Delta City Group (116°41′–117°58′ E, 30°57′–32°32′ N; Figure 1). It has a humid subtropical monsoon climate with an average annual temperature of 15.7 °C and an average annual rainfall of approximately 1000 mm.

The study area includes hills, low mountains, and low-lying plains. The entire area is dominated by hills between the Yangtze and Huai Rivers. The main soil types are anthrosols and luvisols, accounting for approximately 85% of all soil types. The soil profile has a good structure and high nutrient content.

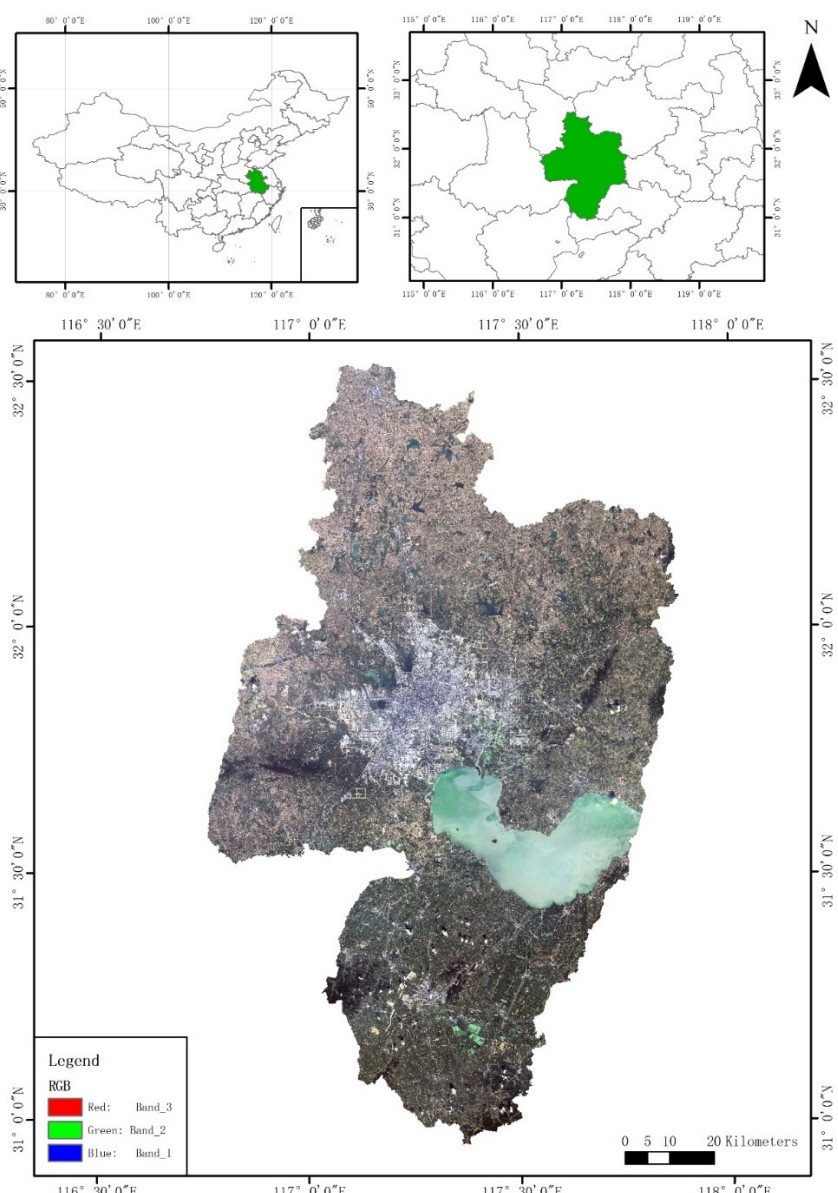

**Figure 1.** Location of the study area.

With a total area of 11,445 km$^2$ and an arable land area of 562,900 ha, Hefei is an important agricultural area in China. Cultivated land is mainly planted with grain crops, with the rotation of grain and non-food crops. The northern part of the study area is mainly planted with wheat, and the cultivated land in the western part of the study area uses a rotation of wheat and commercial forests.

*2.2. Soil Sample Source*

In October 2018, we sampled the cultivated soil in the study area, and a global positioning system (GPS) was used to record the coordinate information of the soil samples (Figure 2). According to the requirements of soil sampling point layout in DZ/T 0295–2016 Specification of Land Quality Geochemical Assessment, 295 topsoil samples were randomly arranged. We used soil auger to sample 0–20 cm soil column samples from the ground

surface. About three to five subsoil columns were collected within a radius of 10 m around the sampling points to form one sample. The 295 soil samples were air-dried, ground, and subjected to other pretreatments, and the organic matter content of the soil was determined by the potassium dichromate-external heating method [57].

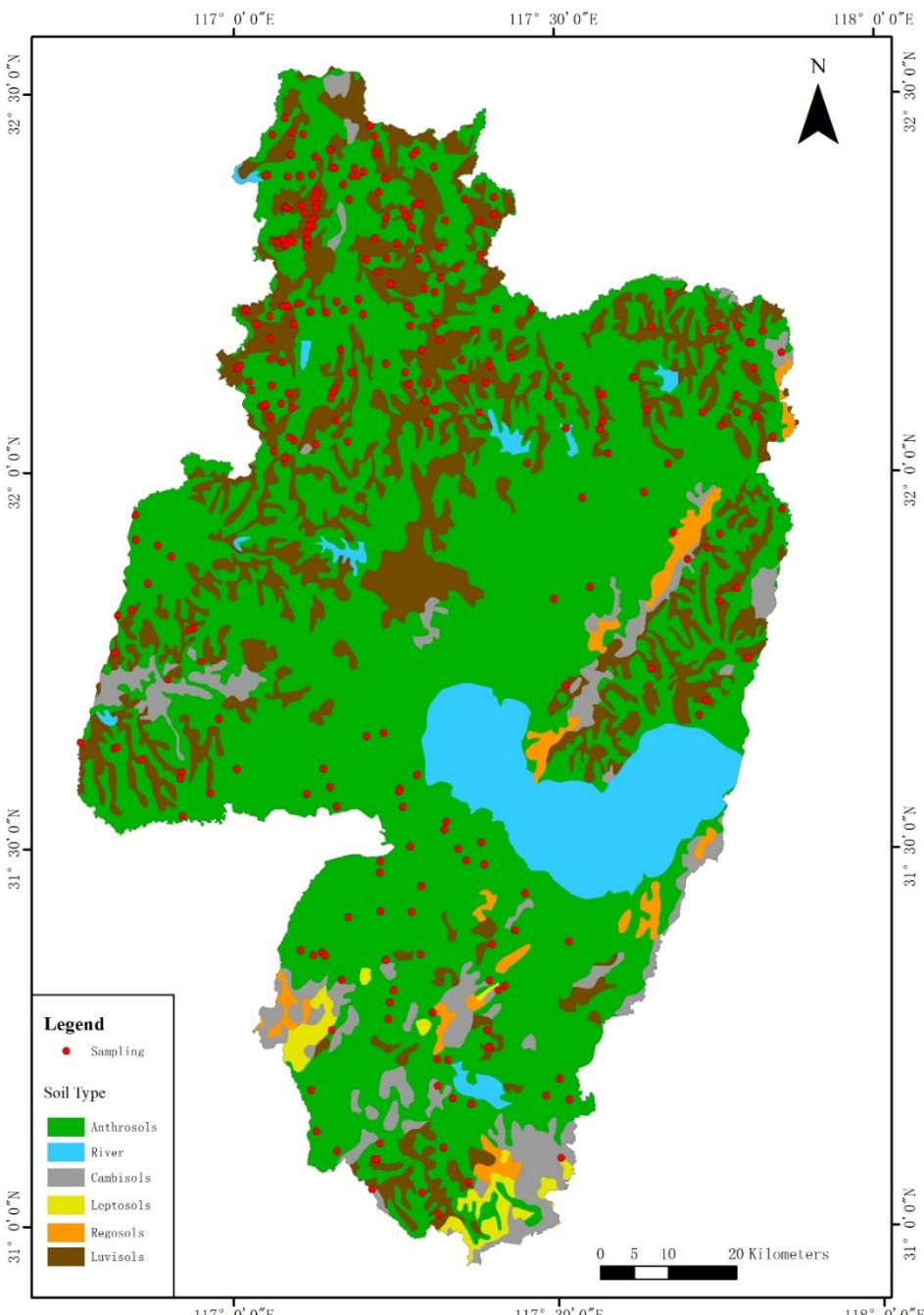

**Figure 2.** Distribution of sampling points on the soil type map.

### 2.3. Remote Sensing Data Processing

2.3.1. GF-6 Image Processing

Launched in June 2018, Gaofen-6 (GF-6) was China's first high-resolution satellite for precision agricultural observations. It has a high resolution, wide coverage, and high localization rate. In this study, we used GF-6 as the data source for obtaining multispectral (MUX) and panchromatic (PAN) data with less than 10% cloud cover at the beginning of October 2018. In ENVI 5.3, we processed the remote sensing image data by radiometric



calibration and atmospheric correction, and the apparent reflectance values were converted into ground reflectance values. The reflectance values of four bands (B1, B2, B3, and B4) of the soil sample points were extracted using the ArcGIS 10.6 software platform (Table 1).

**Table 1.** Gaofen-6 (GF-6) band parameter information.

| Band | Range (μm) |
| --- | --- |
| B1 | 0.45–0.52 |
| B2 | 0.52–0.60 |
| B3 | 0.63–0.69 |
| B4 | 0.76–0.90 |

### 2.3.2. Annual Maximum Synthetic Data

The Google Earth Engine (GEE) platform is an online computing platform developed by Google for macro-scale remote sensing and analysis. Combining the massive remote sensing image data and powerful computing capabilities of the GEE platform, we examined the Landsat8 image dataset from January to December 2018 and from January to December 2019. On this basis, we used the GEE synthesis algorithm to obtain the following maximum synthetic vegetation indices [58] annually pixel by pixel in the study area: the normalized difference vegetation index (NDVI), ratio vegetation index (RVI), difference vegetation index (DVI), and 2018 normalized difference water index (NDWI). As the input feature of the model, we extracted the maximum synthetic vegetation index of the ground sample points.

### 2.3.3. Terrain Data

The 30 m resolution Shuttle Radar Topography Mission (SRTM) image data provided by the GEE platform were used to extract the sampling point elevation (digital elevation model (DEM)) and slope data [59].

### 2.4. Soil Type and Cultivated Land Planting Situation

The soil type data in the study area were obtained from the soil database of the Department of Agriculture and Rural Affairs of Anhui Province. The soil types in the study area are primarily anthrosols and luvisols. Cultivated land planting data were obtained from the third national land survey database of the Department of Natural Resources of Anhui Province. Cultivated land refers to dry land, paddy field, and irrigated land. The cultivated land-planting situation of the sample points in the study area is largely divided into planting food crops and rotation with non-food crops. Food crop in the study area refers to rice, wheat, and corn. Non-food crops in the study area refer to vegetables and fruits.

### 2.5. Spatial Distance Data

Human life and production activities have a long-term impact on the fertility of cultivated land [60,61]. The straight-line distance from the sampling point to the nearest residential area was calculated using the ArcGIS platform as a feature of SOM inversion (Table 2).

### 2.6. Model Building and Testing

As the ground truth data, we selected the soil samples collected in the study area in October 2018, and the GF-6 satellite image data from 4 October 2018 were the remote sensing data. We used Python-based random forest, LightGBM, GBDT, and XGBoost algorithms to establish an inversion model for SOM and remote sensing data. We used the coefficient of determination ($R^2$), root mean square error (RMSE), mean absolute error (MAE), and runtime to evaluate the prediction model.

**Table 2.** Names and descriptions of the datasets used in this study.

| Name of Dataset | Dataset | Resolution | Source |
|---|---|---|---|
| Soil reflectance data | B1-B4 | 8 m | GF-6 |
| Google Earth Engine (GEE) maximum annual indices | 2018NDWIx | 8 m | Landsat8 |
| | 2018NDVIx | 8 m | |
| | 2019NDVIx | 8 m | |
| | 2018DVIx | 8 m | |
| | 2019DVIx | 8 m | |
| | 2018RVIx | 8 m | |
| | 2019RVIx | 8 m | |
| GEE digital elevation model (DEM) data | DEM | 30 m | Shuttle Radar Topography Mission (SRTM) |
| | Slope | 30 m | |
| Soil and cultivated land planting situation | Soil type | Vector data | Department of agriculture |
| | cultivated land planting situation | Vector data | |
| Geostatistical data | Distance | Vector data | Department of natural resources |

### 2.6.1. Model Introduction

Python-based machine learning models.

1.  RF model: The RF algorithm is an ensemble learning method proposed by Breiman in 2001 [62]. The model is a bagged algorithm that contains multiple decision trees. The performance of a single regression tree is improved by combining multiple decision trees. The model output is the result of the integration of multiple decision trees.

2.  GBDT model: The GBDT model is a combination of decision tree and boosting algorithms and was proposed by Friedman in 2001 [63]. The model is an integrated tree model that calculates the residuals between the actual and predicted values. The integrated algorithm model uses gradient, boosting, and decision tree to solve the classification problem and perform the regression prediction. Boosting refers to the offline combination of multiple weak classifiers to achieve a strong classifier, and gradient refers to the increase in flexibility and convenience when the model solves the loss function. Compared to the support vector machine model, the GBDT algorithm has fewer model parameters, faster calculation speed, and higher stability.

3.  LightGBM model: As a part of the GBDT algorithm framework, LightGBM [64] internally integrates multiple decision trees and can integrate the decision results of multiple decision trees, avoiding the low accuracy shortcoming of the use of a single learning machine. The LightGBM algorithm adopts a leaf-wise growth strategy based on histograms, depth limitations, and exclusive feature bundling to increase the speed of calculation and improve training efficiency.

4.  XGBoost model: Chen and Guestrin proposed a new machine learning algorithm called the XGBoost algorithm in 2016 [65]. It has achieved excellent results in many international data mining competitions, and its performance exceeds that of deep learning algorithms [66,67]. The XGBoost algorithm improved on the GBDT algorithm. The loss function is determined by a second-order Taylor expansion, and the regularization concept of the loss function is introduced. The number of constrained nodes and outputs are added to the loss function, which makes the XGBoost algorithm more accurate than the GBDT algorithm, and the algorithm is hard to overfit.

### 2.6.2. Model Evaluation and Tuning the Hyper-Parameters

The sampling points were divided into 265 points (90%) for the training set and 30 points (10%) for the test set for modeling and testing. $R^2$, RMSE, and MAE were used to evaluate the accuracy and stability of the model. K-fold cross-validation was used to validate the training set of the model, and the hyperparameters were adjusted using a grid

search [68,69]. The K-fold cross-validation [70,71] divided the training set into K equal parts, leaving one part as the test set and the rest as the training set. The cross-validation was repeated K times, each sub-sample was verified once, and the average of the K results was taken as the model result. In this study, K = 10 means that we performed 10-fold cross-validation (Figure 3).

$$R^2 = \frac{\left[\sum_{i=1}^{n}\left(C_{ti} - \overline{C}_t\right)\cdot\left(C_{pi} - \overline{C}_p\right)\right]^2}{\sum_{i=1}^{n}\left(C_{ti} - \overline{C}_t\right)^2 \cdot \sum_{i=1}^{n}\left(C_{pi} - \overline{C}_p\right)^2} \tag{1}$$

$$RMSE = \sqrt{\frac{\sum_{i=1}^{n}\left(C_{ti} - C_{pi}\right)^2}{n}} \tag{2}$$

$$MAE = \frac{1}{n}\sum_{i=1}^{n}\left|C_{ti} - C_{pi}\right| \tag{3}$$

where $n$ is the number of samples, $C_{ti}$ is the true SOM content at the $i$-th sample point (g/kg), $C_{pi}$ is the predicted SOM content at the $i$-th sample point (g/kg), $\overline{C}_t$ is the prediction of a true value, and $\overline{C}_p$ is the predicted value.

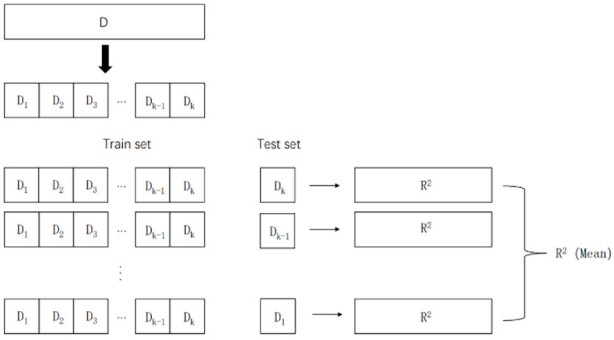

**Figure 3.** K-fold cross validation.

## 3. Results

### 3.1. Statistical Analysis of Soil Organic Matter Content

The soil types used in this study are listed in Table 3. Of the 295 total sampling points, 204 samples were taken from anthrosols (54.86% of the area) and 84 samples were taken from the luvisols (13.90% of the area). The other soil types covered smaller areas and consequently fewer sample points, without suitable conditions for modeling.

**Table 3.** Soil organic matter (SOM) content (g/kg) statistics for all samples, anthrosols samples, and luvisols samples.

| Sampling Dataset | N | SOM | | | | | | |
|---|---|---|---|---|---|---|---|---|
| | | Max | Min | Mean | Standard Deviation | Kurtosis | Skewness | Coefficient of Variation |
| Whole sampling | 295 | 44.6 | 9.8 | 23.19 | 5.894 | 0.384 | 0.048 | 0.254 |
| Anthrosols | 204 | 44.6 | 9.8 | 23.45 | 6.023 | 0.578 | 0.145 | 0.257 |
| Luvisols | 84 | 33.5 | 9.9 | 22.52 | 5.690 | −0.477 | −0.230 | 0.253 |

N = number.

Anthrosols is a type of soil that is strongly influenced by human cultivation, with a high organic matter content [72], with an average of 23.45 g/kg (Figure 4). Luvisols is leached soil developed on the parent material of Pleistocene loess in the fourth quarter [73], with an average organic matter content of 22.52 g/kg. Under the influence of different cultivated land planting situations, we found that the average SOM content of the crop area

was 23.58 g/kg, which is greater than that of the crop rotation area at 22.60 g/kg (Figure 5). The *p*-value of the total sample dataset is 0.0077, which approaches zero, and the dataset tends to be normally distributed.

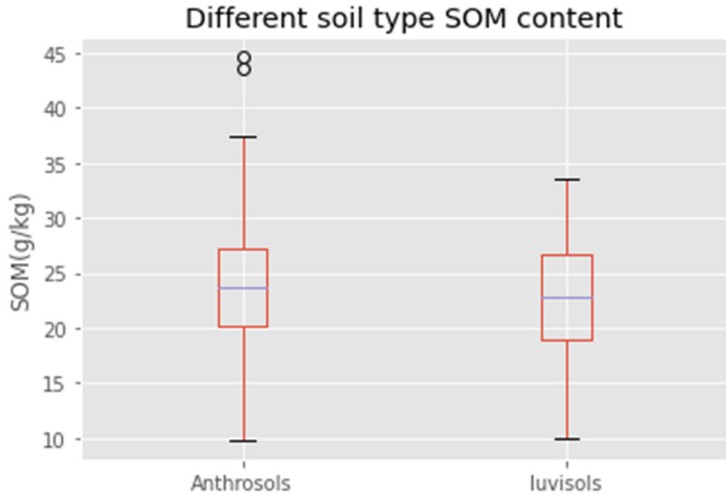

**Figure 4.** Box plot of SOM content in different soil types.

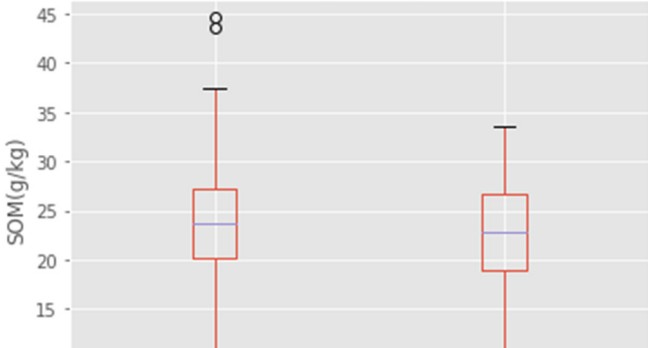

**Figure 5.** Box plot of SOM content in different cultivated land planting situations.

*3.2. Correlation Analysis*

As the features to establish a correlation heat map with SOM content, we used the GF-6 satellite's four-band reflectance data (B1–B4), six synthetic maximum vegetation indices in 2018 and 2019 (2018NDVIx-2019RVIx), 2018 synthetic maximum normalized water index data (2018NDWIx), DEM and slope data, land use data (cultivated land planting situation and soil type), and the spatial distance data between sampling points and residential areas (Figure 6). We concluded that the SOM content of the sampling point is highly correlated with the reflectance data of the four GF-6 bands.

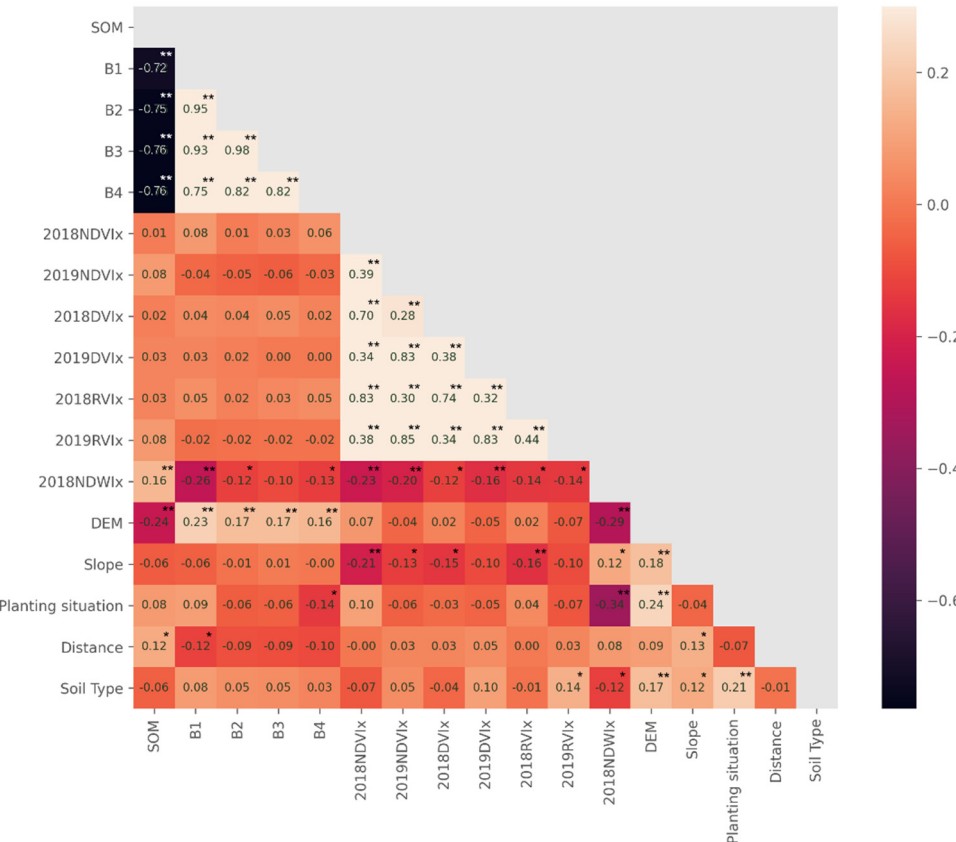

**Figure 6.** Correlation heat map between SOM and features of the study area (* $p \leq 0.05$; ** $p < 0.01$).

### 3.3. Prediction Results of Surface Soil Organic Matter Content

3.3.1. Model Hyperparameter Selection

We built the RF, GBDT, LightGBM, and XGBoost models in Python. A 10-fold cross-validation mean is used to evaluate the training set of the model. After using the default hyperparameters as inputs to obtain the initial results, we adjusted and optimized the corresponding hyperparameters of each model using GridSearchCV [74] to improve the accuracy of the test set of the model (Table 4).

**Table 4.** The optimal parameter, the default parameters, and the range of the grid search in each model.

| Regression Model | Hyperparameters | Optimal Parameter | Default Parameters | Range of Grid Search |
|---|---|---|---|---|
| Extreme gradient boosting machine (XGBoost) | random_state | 0 | 0 | 0 |
| | n_estimators | 15 | 100 | 0~100 |
| | max_depth | 2 | 6 | 1~10 |
| | learning_rate | 0.38 | 0.3 | 0.01~1.00 |
| | min_child_weight | 4 | 1 | 1~10 |
| | gamma | 0.1 | 0 | 0~1.0 |
| Light gradient boosting machine (LightGBM) | random_state | 0 | None | 0 |
| | n_estimators | 26 | 100 | 0~100 |
| | max_depth | 7 | −1 | 1~10 |
| | learning_rate | 0.1 | 0.1 | 0.01~1.00 |
| | subsample | 0.1 | 1 | 0~1.0 |
| Gradient boosting tree (GBDT) | random_state | 0 | None | 0 |
| | n_estimators | 21 | 100 | 0~100 |
| | max_depth | 4 | 3 | 1~10 |
| | learning_rate | 0.18 | 0.1 | 0.01~1.00 |

**Table 4.** *Cont.*

| Regression Model | Hyperparameters | Optimal Parameter | Default Parameters | Range of Grid Search |
|---|---|---|---|---|
| Random Forest (RF) | random_state | 0 | None | 0 |
| | n_estimators | 83 | 100 | 0~100 |
| | max_depth | 8 | None | 1~10 |
| | min_samples_split | 9 | 2 | 1~10 |
| | min_samples_leaf | 1 | 1 | 1~10 |

### 3.3.2. Evaluation of Model Calculation Speed

Our experimental computer is based on Windows 10 system, Core i7 10710 processor, 16G RAM. Different machine learning models exhibit different computing speeds. We compared the calculation speed of different models by comparing the calculation completion time of different models on the same computer platform. We found that RF model was the slowest of the four machine learning models, and the LightGBM model was the fastest computing model (Table 5).

**Table 5.** Runtime of each model.

| Regression Model | Runtime (s) |
|---|---|
| XGBoost | 0.2 |
| LightGBM | 0.1 |
| GBDT | 0.3 |
| RF | 1.4 |

### 3.3.3. Model Accuracy Comparison

We used different algorithms to build models for all 16 features, and we evaluated the accuracy of the models by calculating the $R^2$, RMSE, and MAE values of the training and test sets (Table 6). The XGBoost model had the highest prediction accuracy ($R^2$ = 0.634, RMSE = 3.250, and MAE = 2.637), and the RF model had the lowest accuracy ($R^2$ = 0.551, RMSE = 3.591, and MAE = 2.698).

**Table 6.** Model prediction accuracy.

| Regression Model | Performance Indicator | | |
|---|---|---|---|
| | Coefficient of Determination ($R^2$) | Root Mean Square Error (RMSE) | Mean Absolute Error (MAE) |
| XGBoost | 0.634 | 3.250 | 2.637 |
| LightGBM | 0.627 | 3.278 | 2.618 |
| GBDT | 0.591 | 3.432 | 2.780 |
| RF | 0.551 | 3.591 | 2.698 |

### *3.4. SOM Prediction Results of Different Datasets*

#### 3.4.1. Anthrosols Prediction Result

To explore the role of the different datasets in SOM prediction, samples of anthrosols were selected from the total dataset to form a anthrosols dataset. A 10-fold cross-validation was used to adjust the hyperparameters in the grid search of the anthrosols dataset (Table 7).

**Table 7.** Hyperparameter values for each model in the anthrosols dataset.

| Regression Model | Hyperparameters | Optimal Parameter |
|---|---|---|
| XGBoost | random_state | 0 |
| | n_estimators | 14 |
| | max_depth | 2 |
| LightGBM | random_state | 0 |
| | n_estimators | 60 |
| | max_depth | 6 |
| | learning_rate | 0.08 |
| | subsample | 0.01 |
| GBDT | random_state | 0 |
| | n_estimators | 27 |
| | max_depth | 3 |
| | learning_rate | 0.11 |
| RF | random_state | 0 |
| | n_estimators | 89 |
| | max_depth | 9 |
| | min_samples_split | 3 |
| | min_samples_leaf | 1 |

The XGBoost model(see Appendix A) best fits the anthrosols dataset (Table 8). The $R^2$ of the XGBoost model for anthrosols (0.748) was larger than that for the total dataset (0.634). Among the four machine learning models, the XGBoost, LightGBM, and GBDT models had the same operation speed, and the RF model had the slowest speed. Overall, the model had a faster calculation speed due to the smaller sample size.

**Table 8.** Prediction accuracy and runtime of each model for the anthrosols dataset.

| Regression Model | Runtimes (s) | Performance Indicator | | |
|---|---|---|---|---|
| | | $R^2$ | RMSE | MAE |
| XGBoost | 0.2 | 0.748 | 1.858 | 1.405 |
| LightGBM | 0.2 | 0.355 | 2.974 | 2.198 |
| GBDT | 0.2 | 0.711 | 1.990 | 1.583 |
| RF | 1.4 | 0.741 | 1.880 | 1.649 |

3.4.2. Feature Selection for Anthrosols Dataset

Feature importance is an important reference when selecting features. The XGBoost model uses the gain criterion to calculate the importance of each feature when participating in model training. The gain is calculated by the contribution of the feature to each tree, that is, the contribution of each feature to the generative model. The higher the value, the greater the importance of this feature to the prediction of the model [75]. The importance of each feature in the anthrosols dataset on the XGBoost model was ranked to explore the influence of different features in the same dataset on the prediction accuracy of the model (Figure 7).

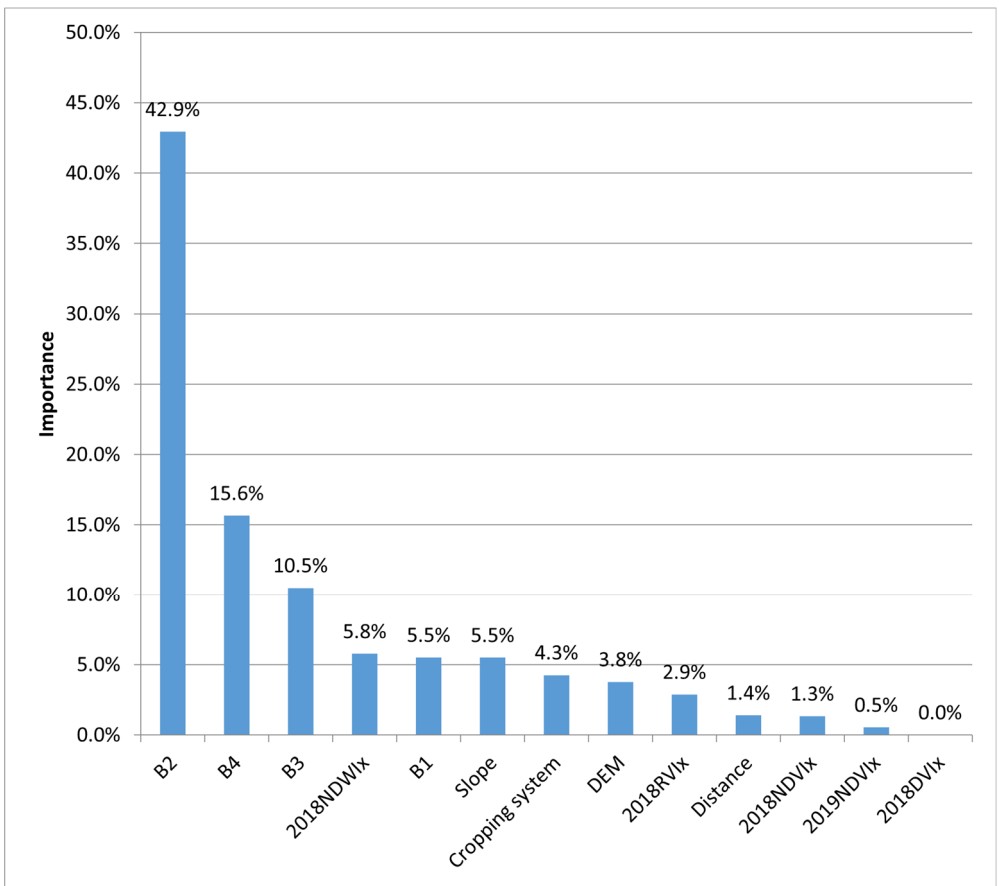

**Figure 7.** Feature importance ranking of the XGBoost model; the sum of importance of all features is 100%.

A 10-fold cross-validation mean was used to evaluate the influence of different numbers of features on the predictive ability of the model. From the experiment, we found that when the number of features involved in the fitting is nine, the maximum cross-validation mean is 0.565. Therefore, we took the top nine features with the most important features to participate in the training, and the model has the best fit (Figure 8).

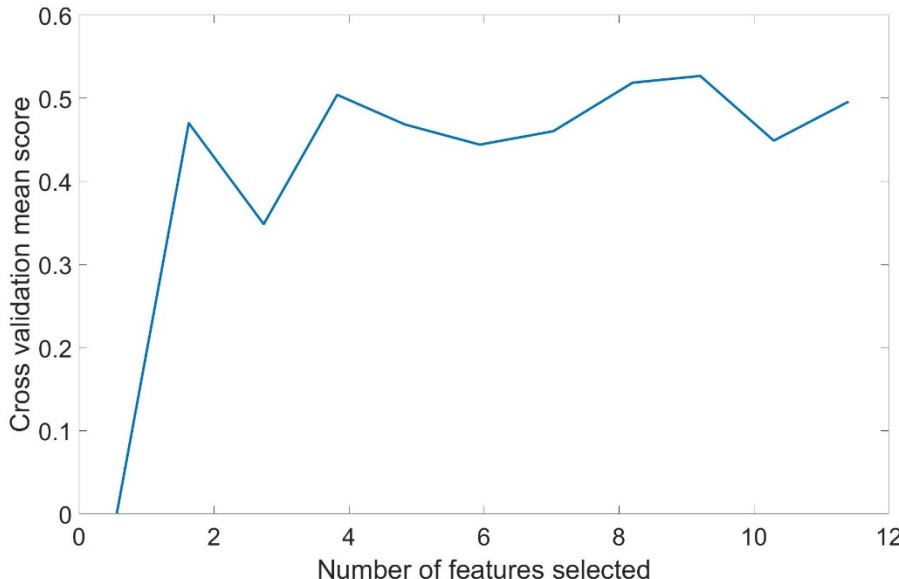

**Figure 8.** Influence of the number of features on the model cross-validation mean.



We selected the top nine features that had the most impact on the model, and we adjusted the hyperparameters of the XGBoost model through a grid search (random_state = 0, n_estimators = 20, and max_depth = 4).

Based on the prediction accuracy and efficiency of the four machine learning models, we found that the XGBoost model has a high accuracy, short runtime, and few hyperparameters that need to be adjusted, making it the most suitable for predicting the spatial distribution of SOM in the research area. After reducing the number of features, the predictive ability of the model was further improved, and $R^2$ reached 0.771. Thus, the predictive ability of the model can be improved by optimizing the number of features (Table 9).

**Table 9.** Anthrosols prediction accuracy and runtime of XGBoost model after feature optimization.

| Regression Model | Runtime (s) | Performance Indicator | | |
|---|---|---|---|---|
| | | $R^2$ | RMSE | MAE |
| XGBoost | 0.2 | 0.771 | 1.773 | 1.474 |

### 3.5. Simulation of the Spatial Distribution of SOM

The XGBoost model trained on the anthrosols dataset after feature selection had the best prediction effect; therefore, we used this model to predict the spatial distribution of SOM content in the study area. Using Python to write a program on the Visual Studio Code platform, the data of the nine features per pixel were transferred to the trained model, and the pixel-by-pixel organic matter content was output to obtain the spatial distribution of the SOM content in the study area (Figure 9).

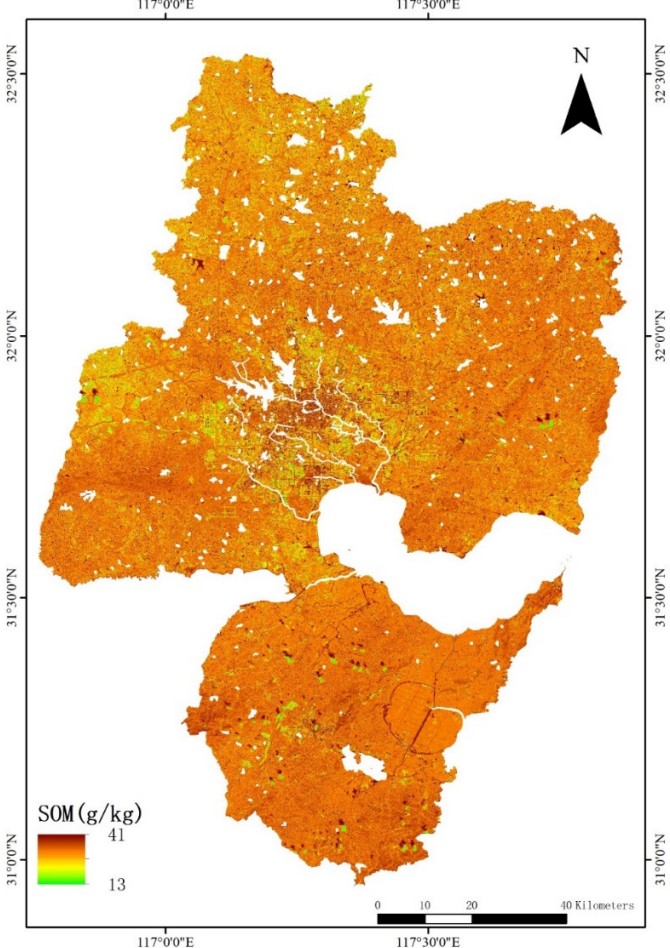

**Figure 9.** Prediction mapping of SOM content in the study area.

## 4. Discussion

### 4.1. Predictive Performance Analysis

In this study, we separately trained four machine learning models. From the observed performance of the model and the results of the training time, most hyperparameters need to be adjusted for the RF model, and the model's prediction accuracy and training time performance are the worst. LightGBM has the fastest training time, but it has a certain dataset, and its stability is low. XGBoost improves the GBDT model at the algorithm level, and its prediction accuracy and efficiency are higher than those of the GBDT model. As such, we found that the XGBoost model performs the best of the considered models.

The overall anthrosols prediction performance results indicate that the differences in the soil type impacts the prediction performance of the model. The reasons for this impact are as follows: (1) Anthrosols are the most widespread in the study area. The soil fertility is greatly affected by humans. The study area began to implement formula fertilization by soil testing policy in 2006. Government organized researchers to fertilize the cultivated land in the study area by measuring the physical and chemical composition of the soil (http://nync.ah.gov.cn/ (accessed on 8 December 2021)). Due to the high proportion of anthrosols cultivated land in the study area (54.86% of the area), regular water and fertilizer adjustments make anthrosols more fertile than other soil types, and the spatial variability of the SOM content is low [76,77]. (2) In a similar study, researchers found that different land use types have different prediction accuracies of SOM content. By comparing the prediction accuracy of dryland and paddy fields, they proposed that the higher coefficient of variation of the dryland data set and the coefficient of variation of the paddy field data set may lead to better prediction accuracy of the dryland data set [78]. In this study, the coefficient of variation for the organic matter content of the anthrosols dataset was greater than that of the total data set. An increase in the coefficient of variation may improve the prediction accuracy of the model.

### 4.2. Selection and Optimization Analysis of Multivariate Data Features

The total dataset used in this study has 16 features, and the nine features that were the most important to the model were selected in the final model feature optimization stage. The reflectance data of four bands of GF-6 (B1–B4), three annual maximum vegetation indices synthesized by GEE (2019RVIx, 2018RVIx, 2019DVIx), cultivated land planting situation data, and DEM data were analyzed. The two largest vegetation indices in 2019 significantly influenced the prediction results of the model. The maximum vegetation index synthesized by GEE may be useful in future soil attribute prediction studies. The cultivated land planting situations were divided into planting food crops and the rotation of food and non-food crops. The average SOM content of cultivated land for food crops is higher than that of cultivated land for rotation, and there are fewer abnormal values. This may be due to the organic fertilizer replacement policy implemented in the study area in 2016 (http://nync.ah.gov.cn/ (accessed on 8 December 2021)). Farmers in the study area are required to use organic fertilizers instead of part of the chemical fertilizers applied to the cultivated land where food crops are grown. This policy started in 2016, replacing chemical fertilizers with organic fertilizers, which can increase the SOM content [79]. Additionally, the altitude of the soil has a substantial impact on SOM, the correlation between elevation and SOM is $-0.24$, and the effect of elevation on SOM has also been confirmed in previous studies [80,81].

### 4.3. GF-6 Modeling Advantage

Researchers primarily use Landsat [21,27,36–38,58,82] and sentinel satellites [20,22,47,68,78] for the inversion of SOM content. Compared to the above multispectral and hyperspectral satellites, the use of GF-6 as the remote sensing image data source in this study has the following advantages: (1) Multispectral satellite data are more suitable for the inversion of SOM content than hyperspectral satellite data. Due to the large number of reflection bands of hyperspectral data, it is necessary to eliminate redundant and cluttered bands with low

correlation before establishing a regression model [50], and (2) GF-6 was the first high-resolution satellite developed by China for precision agriculture. Using GF-6 to predict the distribution of SOM content provides a basis for the development of precision agriculture policies. The province where the study area is located has been actively carrying out the construction of well-facilitated farmland [83] since 2015, and clarifying the SOM content of the cultivated land is of great significance to the construction of well-facilitated farmland (http://nync.ah.gov.cn/ (accessed on 8 December 2021)). Well-facilitated farmland refers to farmland with leveled land, concentrated contiguous areas, complete facilities, supporting farmland, fertile soil, good ecology, strong disaster resistance, is compatible with modern agricultural production and management methods that are suitable for droughts and floods, and is a high and stable yield farmland. The results of this study can be used to monitor the SOM content of farmland when constructing and evaluating well-facilitated farmland.

### 4.4. Model Efficiency Analysis

Traditional SOM inversion methods typically use single-reflectivity mathematical transformations [38,41,50], such as reciprocal transformation, exponential transformation, and log transformation, as the inputs of the model. Such models have low accuracy and slow computational speed [38,50]. The RF machine learning model is unsuitable for future large-scale organic matter inversion research, and XGBoost models will gradually become mainstream in the future of soil property inversion.

### 4.5. Limitations of the Study

The SOM prediction based on a single soil type has limitations, which are mainly related to the soil type and topography of the specific research area. The soil sampling and remote sensing image selection in this study are all after the autumn harvest in this study area. The SOM prediction study was carried out during the bare soil period of the cultivated land, and there was no vegetation affecting the spectral reflectance of the soil. Further work is needed to prove whether this model can be used for SOM prediction in different regions and at different time periods.

### 4.6. Model Selection

Our research used the tree-based algorithm of machine learning. Tree-based algorithm models perform well in predicting linear problems, and researchers can easily interpret the model's predictions from the perspective of input features [44–47]. Deep learning models perform very well in the research of large data sets [84,85], but it is very difficult for researchers to explain the performance of the models in terms of their input features and model parameters [86]. In future research, we will try to use deep learning models to solve the problem of predicting the spatial distribution of SOM content.

## 5. Conclusions

In this study, we used GF-6 satellite, terrain, and soil type data and combined these data with actual ground measurement data to build a remote sensing monitoring method for SOM content based on XGBoost machine learning. We drew the following conclusions:

(1)   By comparing the $R^2$, RMSE, and MAE values of each model, the XGBoost model was found to be the most suitable for predicting the spatial distribution of SOM in the study area. The $R^2$, RMSE, and MAE values of the XGBoost model based on the optimized anthrosols dataset were 0.771, 1.773, and 1.474, respectively.

(2)   In terms of operating efficiency, the run times of the XGBoost, LightGBM, and GBDT models were shorter than those of the traditional RF model.

(3)   Machine learning methods such as XGBoost can achieve rapid and economical inversion of SOM content, allowing their application in precision agriculture.

**Author Contributions:** Resources, Y.M. and Z.L.; software, Z.Y., R.W. and S.D.; visualization, Z.S., M.L. and X.L.; writing, Z.Y. and M.L.; supervision, Q.W. All authors have read and agreed to the published version of the manuscript.

**Funding:** This research was funded by the National Natural Science Foundation of China: (Grant number: 41801234), and the Collaborative Innovation Project of Anhui Provincial Department of Education (No. GXXT-2019-047).

**Institutional Review Board Statement:** Not applicable.

**Informed Consent Statement:** Not applicable.

**Data Availability Statement:** Google Earth Engine (GEE) maximum annual indices and DEM data are publicly available online (https://earthengine.google.com/ (accessed on 8 December 2021)). Soil sample data were obtained from Department of Agriculture and Rural Affairs of Anhui Province.

**Acknowledgments:** We thank the National Natural Science Foundation of China: (Grant number: 41801234), and the Collaborative Innovation Project of Anhui Provincial Department of Education (No. GXXT-2019-047) for their support. We are also grateful to the editor and the reviewers for their helpful comments.

**Conflicts of Interest:** The authors declare no conflict of interest.

## Appendix A

XGBoost prediction model based on the anthrosols dataset.

```
import pandas as pd
import xgboost as xgb
import numpy as np
from sklearn.model_selection import cross_val_score
from sklearn.metrics import mean_squared_error
from sklearn.preprocessing import OneHotEncoder
from sklearn.model_selection import train_test_split

data = pd.read_csv('paddy.csv')

#one-hot enconder
x_onehot = data['soil']
values = np.array(x_onehot)
x_one_hot = OneHotEncoder(sparse = False)
values_re = values.reshape(-1,1)
ohe = x_one_hot.fit_transform(values_re)
ohe_array = np.array(ohe)
ohe_column = pd.DataFrame(ohe_array)
ohe_column.columns = x_one_hot.get_feature_names()
data_drop = data.drop(['soil'], axis = 1)
data_join_l = data_drop.join(ohe_column)
data_join = data_join_l

##split train and test set
group1_train_features = ['b1', 'b2', 'b3', 'b4', '2018max','2019max', 'ZZSXMC', 'dist',
'dvi2018x', 'ndwi2018x', 'rvi2018x', 'dem','hf_slope', 'DVI2019x', 'RVI2019x']
label = ['som']
x = data[group1_train_features]
y = data[label]
x_train,x_test,y_train,y_test = train_test_split(x,y,train_size = 0.9, random_state = 0)

x = data[['b1','b2','b3','b4']]
```

```
y = data['som']

x_train,x_test,y_train,y_test = train_test_split(x,y,test_size = 0.23, random_state = 0)

##xgb regression
model = xgb.XGBRegressor(random_state = 0, n_estimators = 14, made_depth = 2)
model.fit(x_train,y_train)

r_sq=model.score(x_test, y_test)
y_pre = model.predict(x_test)

def MSE(y, y_pre):
    return np.mean((y − y_pre) ** 2)
def RMSE(y, y_pre):
    return np.sqrt(MSE(y, y_pre))
def R2(y, y_pre):
    u = np.sum((y − y_pre) ** 2)
    v = np.sum((y − np.mean(y)) ** 2)
    return 1 − (u/v)

##print
print('r2',r_sq)
print('RMSE',RMSE(y_test,y_pre))
print('MSE',MSE(y_test,y_pre))
```

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
