# Peer review of "Using Machine Learning Algorithms Based on GF-6 and Google Earth Engine to Predict and Map the Spatial Distribution of Soil Organic Matter Content"

_sustainability, doi:10.3390/su132414055_

Round 1
Reviewer 1 Report
The manuscript"Using machine learning algorithms based on GF-6 and Google 2 Earth Engine to predict and map the spatial distribution of soil 3 organic matter content" has tested some machine learning models to assess soil organic matter content. The study is interesting and poses significant novelty. However, some aspects of the paper need further revision. I have made specific comments in the attached file. Here are some general ones:
- I suggest authors state their study hypothesis.
- Soil types should follow international soil classification systems. In the current version, the authors have mixed-up land use and soil type.
- The authors must include, at least, the most important algorithms with respective goodness of fit. If the information set is too large to present within the text, it can be considered as supplementary material.
- Some statistical analyses are not described properly in the method section. Please check the specific comments in the attached document.
- The authors should discuss the limitations and sources of uncertainty of their findings. They should also discuss the temporal (seasonal) variation on ground reflectance and its effect on SOM estimation. Discussion regarding the broader application of the models (forests and other land uses) could strengthen the manuscript.
Author Response
Response to Reviewer 1 Comments
Point 1: I suggest authors state their study hypothesis. 

Response 1: Respected reviewers, we added our research hypothesis at the end of the introduction (line 94 to 100). Please review the following sentences,
In this study, we selected Hefei City, Anhui Province, China as the research area to explore the prediction accuracy of SOM based on multi-source data, such as multi-spectral data, elevation, slope, vegetation index, cultivated land planting situation, and soil type. Our aim was to find the most accurate, fastest, and most stable SOM content inversion method suitable for farmland soil in this research area, to provide a valuable reference for the spatial estimation of SOM content.
Point 2: Soil types should follow international soil classification systems. In the current version, the authors have mixed-up land use and soil type.
Response 2: Respected reviewers, thank you very much for your suggestion. Using the International Soil Classification System will allow more readers to understand the background of soil types in the study area.
As per your suggestion, we used the International Soil Classification System to classify the soil in the study area. According to World Reference Base for Soil Resources, paddy soil should be named anthrosols, and yellow cinnamon soil should be named luvisols. We have also revised the land use information. We changed the cropping system to the cultivated land planting situation to illustrate whether the cultivated land is used to planting food crops or rotate food crops and non-food crops. We have also revised the figures and tables of the full text. Please review the following paragraphs,
2.4. Soil type and cultivated land planting situation
The soil type data in the study area were obtained from the soil database of the Department of Agriculture and Rural Affairs of Anhui Province. The soil types in the study area are primarily anthrosols and luvisols. Cultivated land planting data were obtained from the third national land survey database of the Department of Natural Resources of Anhui Province. The cultivated land planting situation of the sample points in the study area is largely divided into planting food crops and rotation with non-food crops. (line 161-168)
Figure 4. Box plot of SOM content in different soil types.
Figure 2. Distribution of sampling points on the soil type map.
Point 3: The authors must include, at least, the most important algorithms with respective goodness of fit. If the information set is too large to present within the text, it can be considered as supplementary material.

Response 3: Respected reviewers, thank you for your comments on the completeness of the article. Because the code part is too long, we intercepted the code of the XGBoost model prediction anthrosols data set and showed it in Appendix A. Please review the following paragraphs,
Appendix A (line 464-524)
XGBoost prediction model based on the anthrosols dataset.
import pandas as pd
import xgboost as xgb
import numpy as np
from sklearn.model_selection import cross_val_score
from sklearn.metrics import mean_squared_error
from sklearn.preprocessing import OneHotEncoder
from sklearn.model_selection import train_test_split
data=pd.read_csv('paddy.csv')
#one-hot enconder
x_onehot = data['soil']
values = np.array(x_onehot)
x_one_hot = OneHotEncoder(sparse=False)
values_re = values.reshape(-1,1)
ohe = x_one_hot.fit_transform(values_re)
ohe_array = np.array(ohe)
ohe_column = pd.DataFrame(ohe_array)
ohe_column.columns = x_one_hot.get_feature_names()
data_drop = data.drop(['soil'],axis = 1)
data_join_l = data_drop.join(ohe_column)
data_join = data_join_l
##split train and test set
group1_train_features = ['b1', 'b2', 'b3', 'b4', '2018max','2019max', 'ZZSXMC', 'dist', 'dvi2018x', 'ndwi2018x', 'rvi2018x', 'dem','hf_slope', 'DVI2019x', 'RVI2019x']
label = ['som']
x = data[group1_train_features]
y = data[label]
x_train,x_test,y_train,y_test=train_test_split(x,y,train_size = .9,random_state=0)
x=data[['b1','b2','b3','b4']]
y=data['som']
x_train,x_test,y_train,y_test=train_test_split(x,y,test_size=0.23,random_state=0)
##xgb regression
model=xgb.XGBRegressor(random_state=0,n_estimators=14,made_depth=2)
model.fit(x_train,y_train)
r_sq=model.score(x_test,y_test)
y_pre = model.predict(x_test)
def MSE(y, y_pre):
return np.mean((y - y_pre) ** 2)
def RMSE(y, y_pre):
return np.sqrt(MSE(y, y_pre))
def R2(y, y_pre):
u = np.sum((y - y_pre) ** 2)
v = np.sum((y - np.mean(y)) ** 2)
return 1 - (u / v)
print('r2',r_sq)
print('RMSE',RMSE(y_test,y_pre))
print('MSE',MSE(y_test,y_pre))
Point 4: Some statistical analyses are not described properly in the method section. Please check the specific comments in the attached document. 

Response 4: Respected reviewers, thank you for your comments on the statistical analysis, but we couldn't seem to find the attachment you mentioned. We added a P-value test to the correlation of features. We added * to the correlation heat map to show their significance. Please review the following paragraph,
3.2. Correlation Analysis
As the features to establish a correlation heat map with SOM content, we used the GF-6 satellite’s four-band reflectance data (B1–B4), six synthetic maximum vegetation indices in 2018 and 2019 (2018NDVIx-2019RVIx), 2018 synthetic maximum normalized water index data (2018NDWIx), DEM and slope data, land use data (cultivated land planting situation and soil type), and the spatial distance data between sampling points and residential areas(Figure 6). We concluded that the SOM content of the sampling point is highly correlated with the reflectance data of the four GF-6 bands. (line 254-265)
Figure 6. Correlation heat map between SOM and features of the study area(* p ≤ 0.05; ** p < 0.01).
Point 5: The authors should discuss the limitations and sources of uncertainty of their findings. They should also discuss the temporal (seasonal) variation on ground reflectance and its effect on SOM estimation. Discussion regarding the broader application of the models (forests and other land uses) could strengthen the manuscript. 

Response 5: Respected reviewers, thank you very much for your suggestions on the limitations of the article. We are very interested in the question of whether the prediction of SOM content can be extended from the bare soil period of cultivated land to the surface covered by vegetation. How to predict the SOM content of other land use types requires further exploration. We have included research limitations in the discussion section of the manuscript. Please review the following paragraph,
4.5. Limitations of the study
The SOM prediction based on a single soil type has limitations, which are mainly related to the soil type and topography of the specific research area. The soil sampling and remote sensing image selection in this study are all after the autumn harvest in this study area. The SOM prediction study was carried out during the bare soil period of the cultivated land, and there was no vegetation affecting the spectral reflectance of the soil. Further work is needed to prove whether this model can be used for SOM prediction in different regions and at different time periods. (line 428-436)
We tried to improve the manuscript and made some changes in the manuscript. We appreciate for your warm work earnestly, and hope that the correction will meet with approval.
Once again, thank you very much for your comments and suggestions.

Reviewer 2 Report
Overall the manuscript requires an extensive editing of English language and style. From a methodological point of view, the manuscript add some novelty to the field of coupling machine learning to remote sensing data for estimating key soil properties. Discussion section in particular needs to be improved describing how the findings may address and solve the challenges in SOM management.
Specific comments are:
Line 30-33: more emphasis should be put on the role of SOM on supporting ecosystem services and crop productivity
Line 36: please change scholars with scientists or similars.
Line 48 -49: this are obvious sentences with no references. Please rephrase them
Line 57: please describe better what is the SOM inversion method
Line 69-71: references are missing
Line 86: soil and farmland..the SOM content of which soil are going to be predicted ? monitoring fertility and spatial estimation..these are two different topics. The objective is not clear. Please be more precise.
Line 96: which soil taxonomy is ?
Line 106 : how the soil sampling has been conducted ? which soil coring equipment ? soil depth sampled ?
Line 108-110: the samples were 2-mm sieved ? please report the reference for the analytical method.
Line 192: specify the hyperparameters and the grid search used reporting a table with values range, the initial starting combination are not shown.
Line 207: planting method and soil type are not the right terms to describe the sampling dataset of table 3
Line 226 please use USDA or FAO-WRB soil orders taxonomy to describe soil of your region.
Line 226 crop and crop rotation are not cropping systems. Paddy rice, wheat, or more generally cropland, grassland, forest are more appropriate terms to describe land uses.
Line 228-234 are these correlations significant (p<0.05) ?
Line 239: Visual Studio Code is not programming language . I would specify instead the Phyton’s modules that have been used .
Line 240: why adjusted R2 has been used ?
Line 260: it’s a repetition from M&M section
Line 279 – which is the criteria by which nine features is the best number and not 2 ? Please rephrase this sentence.
Figures 7-8 should be both re-drawn with e.g. in Fig. 7 fewer digits after comma in the percentages and fig.8 improved the image’s quality
Line 310 – the use of past tense would be prefereable.
Line 314-317 this sentence does not have scientific relevance neither from an agronomic point of view nor from a modelling one.
Line 320 a sentence with …”similar results in previous studies” does not discuss at all your results. Please contextualize and discuss more in depth your results with the ones found in other studies.
Line 327-328: this is a too general sentence that cannot be used to discuss the fact that vegetational indices had an influence on 2019 predicted results.
Line 328-330: what are planting food crops ? Throughout all manuscript the terminology used for describing cropping systems and relative land use are inconsistent .
Line 331- as above “human adjustment of SOM” is inappropriate. SOM can be managed by farmers with organic fertilization, conservation tillage , targeted crop rotation etc… which are the other factor ? Please be more precise and consistent in result’s discussion.
Line 341 – “a lot of noise” please be more precise
Line 343 – what is “inversion research” ?
Line 345 – not clear. Please rephrase. What do “high-standard farmland” and “follow-up precise management” mean ?
Author Response
Response to Reviewer 2 Comments
Point 1: Line 30-33: more emphasis should be put on the role of SOM on supporting ecosystem services and crop productivity
Response 1: Respected reviewers, thank you very much for your comments. We added the content of predicting soil organic matter (SOM) content to support ecosystem services and crop productivity in the introduction and cited related papers (line 35-37). Please review the following sentence,
Determining the SOM content is vital for achieving sustainable agricultural development [10] and ecological civilization [11], supporting ecosystem services [12] and improving crop productivity[13,14]. (Line 35-37)
Reference:
- Jiang, Z.; Lian, F.; Wang, Z.; Xing, B. The role of biochars in sustainable crop production and soil resiliency. Journal of Experimental Botany 2020, 71, 520-542, doi:10.1093/jxb/erz301.
- Ramesh, T.; Bolan, N.S.; Kirkham, M.B.; Wijesekara, H.; Kanchikerimath, M.; Rao, C.S.; Sandeep, S.; Rinklebe, J.; Ok, Y.S.; Choudhury, B.U., et al. Soil organic carbon dynamics: Impact of land use changes and management practices: A review. In Advances in Agronomy, Vol 156, Sparks, D.L., Ed. 2019; Vol. 156, pp. 1-107.
- Velasquez, E.; Lavelle, P. Soil macrofauna as an indicator for evaluating soil based ecosystem services in agricultural landscapes. ACTA OECOLOGICA-INTERNATIONAL JOURNAL OF ECOLOGY 2019, 100, doi:10.1016/j.actao.2019.103446.
- Oldfield, E.E.; Wood, S.A.; Bradford, M.A. Direct effects of soil organic matter on productivity mirror those observed with organic amendments. Plant Soil 2018, 423, 363-373, doi:10.1007/s11104-017-3513-5.
- Zhao, Y.N.; He, X.H.; Huang, X.C.; Zhang, Y.Q.; Shi, X.J. Increasing Soil Organic Matter Enhances Inherent Soil Productivity while Offsetting Fertilization Effect under a Rice Cropping System. Sustainability 2016, 8, 12, doi:10.3390/su8090879.
Point 2: Line 36: please change scholars with scientists or similars.
Response 2: Respected reviewers, thank you very much for your comments. We have replaced the scholars in the full text with researchers. Please review the following paragraphs,
With the increasing demand for informatization for precision agriculture, accurately, quickly, and extensively estimating the SOM content has become challenging for many researchers. (Line 38-40)
Researchers combine spectral characteristics with SOM content using remote sensing technology. Retrieving SOM content is, thus, a hot topic in soil science research [22-28].(Line 53-55)
Researchers primarily use Landsat [21,27,36-38,58,81] and sentinel satellites [20,22,47,68,78] for the inversion of SOM content. (Line 403-404)
Reference:
- Pouladi, N.; Møller, A.B.; Tabatabai, S.; Greve, M.H. Mapping soil organic matter contents at field level with Cubist, Random Forest and kriging. Geoderma 2019, 342, 85-92, doi:https://doi.org/10.1016/j.geoderma.2019.02.019.
- Henderson, T.L.; Szilagyi, A.; Baumgardner, M.F.; Chen, C.-c.T.; Landgrebe, D.A. Spectral Band Selection for Classification of Soil Organic Matter Content. Soil Science Society of America Journal 1989, 53, 1778-1784, doi:https://doi.org/10.2136/sssaj1989.03615995005300060028x.
- Zhang, M.; Zhang, M.; Yang, H.; Jin, Y.; Zhang, X.; Liu, H. Mapping Regional Soil Organic Matter Based on Sentinel-2A and MODIS Imagery Using Machine Learning Algorithms and Google Earth Engine. REMOTE SENSING 2021, 13, doi:10.3390/rs13152934.
- Santaga, F.S.; Agnelli, A.; Leccese, A.; Vizzari, M. Using Sentinel-2 for Simplifying Soil Sampling and Mapping: Two Case Studies in Umbria, Italy. 2021, 13, 3379.
- Meng, X.; Bao, Y.; Ye, Q.; Liu, H.; Zhang, X.; Tang, H.; Zhang, X. Soil Organic Matter Prediction Model with Satellite Hyperspectral Image Based on Optimized Denoising Method. Remote Sensing 2021, 13, doi:10.3390/rs13122273.
- Nanni, M.R.; Demattê, J.A.; Rodrigues, M.; Santos, G.L.; Reis, A.S.; Oliveira, K.M.; Cezar, E.; Furlanetto, R.H.; Crusiol, L.G.; Sun, L. Mapping Particle Size and Soil Organic Matter in Tropical Soil Based on Hyperspectral Imaging and Non-Imaging Sensors. Remote Sensing 2021, 13, doi:10.3390/rs13091782.
- Gomez, C.; Rossel, R.A.V.; McBratney, A.B. Soil organic carbon prediction by hyperspectral remote sensing and field vis-NIR spectroscopy: An Australian case study. GEODERMA 2008, 146, 403-411, doi:10.1016/j.geoderma.2008.06.011.
- Li, X.-p.; Zhang, F.; Wang, X.-p. Study on Differential-Based Multispectral Modeling of Soil Organic Matter in Ebinur Lake Wetland. SPECTROSCOPY AND SPECTRAL ANALYSIS 2019, 39, 535-542, doi:10.3964/j.issn.1000-0593(2019)02-0535-08.
- Zhai, M. Inversion of organic matter content in wetland soil based on Landsat 8 remote sensing image. JOURNAL OF VISUAL COMMUNICATION AND IMAGE REPRESENTATION 2019, 64, doi:10.1016/j.jvcir.2019.102645.
- Zhang, Y.; Guo, L.; Chen, Y.; Shi, T.; Luo, M.; Ju, Q.; Zhang, H.; Wang, S. Prediction of Soil Organic Carbon based on Landsat 8 Monthly NDVI Data for the Jianghan Plain in Hubei Province, China. Remote Sensing 2019, 11, doi:10.3390/rs11141683.
- Yu, H.; Liu, M.; Du, B.; Wang, Z.; Hu, L.; Zhang, B. Mapping Soil Salinity/Sodicity by using Landsat OLI Imagery and PLSR Algorithm over Semiarid West Jilin Province, China. Sensors 2018, 18, doi:10.3390/s18041048.
- Fu, C.; Gan, S.; Yuan, X.; Xiong, H.; Tian, A. Impact of Fractional Calculus on Correlation Coefficient between Available Potassium and Spectrum Data in Ground Hyperspectral and Landsat 8 Image. Mathematics 2019, 7, doi:10.3390/math7060488.
- Wang, X.; Han, J.; Wang, X.; Yao, H.; Zhang, L. Estimating Soil Organic Matter Content Using Sentinel-2 Imagery by Machine Learning in Shanghai. IEEE ACCESS 2021, 9, 78215-78225, doi:10.1109/ACCESS.2021.3080689.
- Liu, L.; Ji, M.; Buchroithner, M. Combining Partial Least Squares and the Gradient-Boosting Method for Soil Property Retrieval Using Visible Near-Infrared Shortwave Infrared Spectra. REMOTE SENSING 2017, 9, doi:10.3390/rs9121299.
- Wang, Z.; Wang, G.; Zhang, Y.; Wang, R. Quantification of the effect of soil erosion factors on soil nutrients at a small watershed in the Loess Plateau, Northwest China. JOURNAL OF SOILS AND SEDIMENTS 2020, 20, 745-755, doi:10.1007/s11368-019-02458-5.
- Ahirwal, J.; Nath, A.; Brahma, B.; Deb, S.; Sahoo, U.K.; Nath, A.J. Patterns and driving factors of biomass carbon and soil organic carbon stock in the Indian Himalayan region. SCIENCE OF THE TOTAL ENVIRONMENT 2021, 770, doi:10.1016/j.scitotenv.2021.145292.
- Zhou, L.; Luo, T.; Du, M.; Chen, Q.; Liu, Y.; Zhu, Y.; He, C.; Wang, S.; Yang, K. Machine Learning Comparison and Parameter Setting Methods for the Detection of Dump Sites for Construction and Demolition Waste Using the Google Earth Engine. Remote Sensing 2021, 13, doi:10.3390/rs13040787.
- Wang, H.; Zhang, X.; Wu, W.; Liu, H. Prediction of Soil Organic Carbon under Different Land Use Types Using Sentinel-1/-2 Data in a Small Watershed. Remote Sensing 2021, 13, doi:10.3390/rs13071229.
- Tian, F.; Wang, Y.; Fensholt, R.; Wang, K.; Zhang, L.; Huang, Y. Mapping and Evaluation of NDVI Trends from Synthetic Time Series Obtained by Blending Landsat and MODIS Data around a Coalfield on the Loess Plateau. REMOTE SENSING 2013, 5, 4255-4279, doi:10.3390/rs5094255.
- Zhai, M. Inversion of organic matter content in wetland soil based on Landsat 8 remote sensing image. Journal of Visual Communication and Image Representation 2019, 64, 102645.
Point 3: Line 48 -49: this are obvious sentences with no references. Please rephrase them
Response 3: Respected reviewers, thank you very much for your comments. We have added a citation for this sentence. Please review the following sentence,
Researchers have combined spectral characteristics with SOM content using remote sensing technology. Retrieving the SOM content is, thus, a hot topic in soil science research [22-28]. (Line 53-55)
Reference:
- Liu, L.; Ji, M.; Buchroithner, M. Combining Partial Least Squares and the Gradient-Boosting Method for Soil Property Retrieval Using Visible Near-Infrared Shortwave Infrared Spectra. REMOTE SENSING 2017, 9, doi:10.3390/rs9121299.
- Wang, Z.; Wang, G.; Zhang, Y.; Wang, R. Quantification of the effect of soil erosion factors on soil nutrients at a small watershed in the Loess Plateau, Northwest China. JOURNAL OF SOILS AND SEDIMENTS 2020, 20, 745-755, doi:10.1007/s11368-019-02458-5.
- Ahirwal, J.; Nath, A.; Brahma, B.; Deb, S.; Sahoo, U.K.; Nath, A.J. Patterns and driving factors of biomass carbon and soil organic carbon stock in the Indian Himalayan region. SCIENCE OF THE TOTAL ENVIRONMENT 2021, 770, doi:10.1016/j.scitotenv.2021.145292.
Point 4: Line 57: please describe better what is the SOM inversion method
Response 4: Respected reviewers, thank you very much for your comments. We explained the inversion method of SOM content. Please review the following sentences,
The SOM content of soil presented in a remote sensing image can be predicted by a regression model. This Model established a relationship between the spectral reflectance and SOM content of ground samples. When putting the spectral reflectance information of non-sampling points into it, the SOM content is the result which got from the calculation in regression equation. (Line 67-71)
Point 5: Line 69-71: references are missing
Response 5: Respected reviewers, thank you very much for your comments. We added references for this sentence. Please review the following sentence,
Machine learning algorithms have been gradually introduced for the prediction of various soil properties in the fields of mathematics and computers [44-47]. (Line 80-82)
Reference:
- Emadi, M.; Taghizadeh-Mehrjardi, R.; Cherati, A.; Danesh, M.; Mosavi, A.; Scholten, T. Predicting and Mapping of Soil Organic Carbon Using Machine Learning Algorithms in Northern Iran. REMOTE SENSING 2020, 12, doi:10.3390/rs12142234.
- Kobayashi, Y.; Yoshida, K. Quantitative structure?property relationships for the calculation of the soil adsorption coefficient using machine learning algorithms with calculated chemical properties from open-source software. ENVIRONMENTAL RESEARCH 2021, 196, doi:10.1016/j.envres.2020.110363.
- Dong, Z.; Wang, N.; Liu, J.; Xie, J.; Han, J. Combination of machine learning and VIRS for predicting soil organic matter. JOURNAL OF SOILS AND SEDIMENTS 2021, 21, 2578-2588, doi:10.1007/s11368-021-02977-0.
- Wang, X.; Han, J.; Wang, X.; Yao, H.; Zhang, L. Estimating Soil Organic Matter Content Using Sentinel-2 Imagery by Machine Learning in Shanghai. IEEE ACCESS 2021, 9, 78215-78225, doi:10.1109/ACCESS.2021.3080689.
Point 6: Line 86: soil and farmland..the SOM content of which soil are going to be predicted ? monitoring fertility and spatial estimation..these are two different topics. The objective is not clear. Please be more precise.
Response 6: Respected reviewers, thank you very much for your comments. We revised this sentence. Please review the following sentence,
Our aim was to find the most accurate, fastest, and most stable SOM content inversion method suitable for farmland soil in this research area, to provide a valuable reference for the spatial estimation of SOM content. (Line 97-100)
Point 7: Line 96: which soil taxonomy is ?
Response 7: Respected reviewers, thank you very much for your comments. We have revised the soil names in the full text and renamed them using the International Soil Classification System. Using the International Soil Classification System will allow more readers to understand the background of soil types in the study area. Please review the following sentence,
The entire area is dominated by hills between the Yangtze and Huai Rivers. The main soil types are anthrosols and luvisols, accounting for approximately 85% of all soil types. (Line 109-111)
Point 8: Line 106: how the soil sampling has been conducted? which soil coring equipment? soil depth sampled?
Response 8: Respected reviewers, thank you very much for your comments. We have added the content of soil sample collection. We revised this sentence. Please review the following sentences,
According to the requirements of soil sampling point layout in DZ/T 0295–2016 Specification of Land Quality Geochemical Assessment, 295 topsoil samples were randomly arranged. We used soil auger to sample 0-20cm soil column samples from the ground surface. About 3–5 subsoil columns were collected within a radius of 10 m around the sampling points to form one sample. (Line 123-127)
Point 9: Line 108-110: the samples were 2-mm sieved? please report the reference for the analytical method.
Response 9: Respected reviewers, thank you very much for your comments. We have added a reference to the soil sample processing method. We revised this sentence. Please review the following sentence,
The 295 soil samples were air-dried, ground, and subjected to other pretreatments, and the organic matter content of the soil was determined by the potassium dichromate-external heating method [57]. (Line 127-129)
Reference:
- Li, M.; Xi, X.; Xiao, G.; Cheng, H.; Yang, Z.; Zhou, G.; Ye, J.; Li, Z. National multi-purpose regional geochemical survey in China. Journal of Geochemical Exploration 2014, 139, 21-30, doi:10.1016/j.gexplo.2013.06.002.
Point 10: Line 192: specify the hyperparameters and the grid search used reporting a table with values range, the initial starting combination are not shown.
Response 10: Respected reviewers, thank you very much for your comments. The specific value of the hyperparameter optimization search has been added to the table. Please review the following paragraphs,
Table 4. The optimal parameter, the default parameters and the range of the grid search in each model. (Line 275)
|
Regression Model |
Hyperparameters |
Optimal Parameter |
Default Parameters |
Range of Grid Search |
||
|
Extreme gradient boosting machine (XGBoost) |
random_state |
0 |
0 |
0 |
||
|
n_estimators |
15 |
100 |
0~100 |
|||
|
max_depth |
2 |
6 |
1~10 |
|||
|
learning_rate |
0.38 |
0.3 |
0.01~1.00 |
|||
|
min_child_weight |
4 |
1 |
1~10 |
|||
|
gamma |
0.1 |
0 |
0~1.0 |
|||
|
Light gradient boosting machine (LightGBM) |
random_state |
0 |
None |
0 |
||
|
n_estimators |
26 |
100 |
0~100 |
|||
|
max_depth |
7 |
-1 |
1~10 |
|||
|
learning_rate |
0.1 |
0.1 |
0.01~1.00 |
|||
|
subsample |
0.1 |
1 |
0~1.0 |
|||
|
Gradient boosting tree (GBDT) |
random_state |
0 |
None |
0 |
||
|
n_estimators |
21 |
100 |
0~100 |
|||
|
max_depth |
4 |
3 |
1~10 |
|||
|
learning_rate |
0.18 |
0.1 |
0.01~1.00 |
|||
|
Random Forest (RF) |
random_state |
0 |
None |
0 |
||
|
n_estimators |
83 |
100 |
0~100 |
|||
|
max_depth |
8 |
None |
1~10 |
|||
|
min_samples_split |
9 |
2 |
1~10 |
|||
|
min_samples_leaf |
1 |
1 |
1~10 |
|||
Point 11: Line 207: planting method and soil type are not the right terms to describe the sampling dataset of table 3
Response 11: Respected reviewers, thank you very much for your comments. We have replaced the soil names in Table 3 with the names of the International Soil Classification System. Please review the following paragraphs,
Table 3. Soil organic matter (SOM) content (g/kg) statistics for all samples, anthrosols samples, and luvisols samples. (Line 247)
|
Sampling Dataset |
N |
SOM |
|
||||||
|
Max |
Min |
Mean |
Standard Deviation |
Kurtosis |
Skewness |
Coefficient of Variation |
|
||
|
Whole sampling |
295 |
44.6 |
9.8 |
23.19 |
5.894 |
0.384 |
0.048 |
0.254 |
|
|
Anthrosols |
204 |
44.6 |
9.8 |
23.45 |
6.023 |
0.578 |
0.145 |
0.257 |
|
|
Luvisols |
84 |
33.5 |
9.9 |
22.52 |
5.690 |
-0.477 |
-0.230 |
0.253 |
|
N = number
Point 12: Line 226 please use USDA or FAO-WRB soil orders taxonomy to describe soil of your region.
Response 12: Respected reviewers, thank you very much for your comments. We have replaced the soil names in Table 3 with the names of the FAO-WRB soil orders taxonomy. Paddy soil should be named anthrosols, and yellow cinnamon soil should be named luvisols. We revised this sentence. Please review the following paragraph,
Figure 4. Box plot of SOM content in different soil types. (Line 251)
Point 13: Line 226 crop and crop rotation are not cropping systems. Paddy rice, wheat, or more generally cropland, grassland, forest are more appropriate terms to describe land uses.
Response 13: Respected reviewers, thank you very much for your comments. We revised the planting situation of the cultivated land where the sample point was located at the time of sampling. We revised this sentence. Please review the following paragraph,
Figure 5. Box plot of SOM content in different cultivated land planting situation. (Line 253)
Point 14: Line 228-234 are these correlations significant (p<0.05) ?
Response 14: Respected reviewers, thank you very much for your comments. We redraw the correlation heat map and marked the significance. Please review the following paragraph,
Figure 6. Correlation heat map between SOM and features of the study area(* p ≤ 0.05; ** p < 0.01). (Line 265)
Point 15: Line 239: Visual Studio Code is not programming language. I would specify instead the Phyton’s modules that have been used.
Response 15: Respected reviewers, thank you very much for your comments. We revised this sentence. Please review the following sentence,
We built the RF, GBDT, LightGBM, and XGBoost models in Python. (Line 268-269)
Point 16: Line 240: why adjusted R2 has been used?
Response 16: Respected reviewers, thank you very much for your comments. We revised this sentence. Please review the following paragraphs,
A ten-fold cross-validation mean is used to evaluate the training set of the model. After using the default hyperparameters as inputs to obtain the initial results, we adjusted and optimized the corresponding hyperparameters of each model using GridSearchCV [74] to improve the accuracy of the test set of the model (Table 4). (Line 268-274)
Table 4.The optimal parameter, the default parameters and the range of the grid search in each model.
|
Regression Model |
Hyperparameters |
Optimal Parameter |
Default Parameters |
Range of Grid Search |
||
|
Extreme gradient boosting machine (XGBoost) |
random_state |
0 |
0 |
0 |
||
|
n_estimators |
15 |
100 |
0~100 |
|||
|
max_depth |
2 |
6 |
1~10 |
|||
|
learning_rate |
0.38 |
0.3 |
0.01~1.00 |
|||
|
min_child_weight |
4 |
1 |
1~10 |
|||
|
gamma |
0.1 |
0 |
0~1.0 |
|||
|
Light gradient boosting machine (LightGBM) |
random_state |
0 |
None |
0 |
||
|
n_estimators |
26 |
100 |
0~100 |
|||
|
max_depth |
7 |
-1 |
1~10 |
|||
|
learning_rate |
0.1 |
0.1 |
0.01~1.00 |
|||
|
subsample |
0.1 |
1 |
0~1.0 |
|||
|
Gradient boosting tree (GBDT) |
random_state |
0 |
None |
0 |
||
|
n_estimators |
21 |
100 |
0~100 |
|||
|
max_depth |
4 |
3 |
1~10 |
|||
|
learning_rate |
0.18 |
0.1 |
0.01~1.00 |
|||
|
Random Forest (RF) |
random_state |
0 |
None |
0 |
||
|
n_estimators |
83 |
100 |
0~100 |
|||
|
max_depth |
8 |
None |
1~10 |
|||
|
min_samples_split |
9 |
2 |
1~10 |
|||
|
min_samples_leaf |
1 |
1 |
1~10 |
|||
Reference:
- Villamil-Cubillos, L.F.; Leon-Medina, J.X.; Anaya, M.; Tibaduiza, D.A. Evaluation of Feature Selection Techniques in a Multifrequency Large Amplitude Pulse Voltammetric Electronic Tongue. Engineering Proceedings 2020, 2, doi:10.3390/ecsa-7-08242.
Point 17: Line 260: it’s a repetition from M&M section
Response 17: Respected reviewers, thank you very much for your comments. We revised this sentence. Please review the following sentence,
To explore the role of the different datasets in SOM prediction, samples of anthrosols were selected from the total dataset to form an anthrosols dataset. (Line 295-297)
Point 18: Line 279 – which is the criteria by which nine features is the best number and not 2 ? Please rephrase this sentence.
Response 18: Respected reviewers, thank you very much for your comments. We revised this sentence. Please review the following paragraphs,
A 10-fold cross-validation mean was used to evaluate the influence of different numbers of features on the predictive ability of the model. From the experiment, we found that when the number of features involved in the fitting is nine, the maximum cross-validation mean is 0.565. Therefore, we took the top nine features with the most important features to participate in the training, and the model has the best fit. (Figure 8). (Line 320-328)
Figure 8. Influence of the number of features on the model cross-validation mean.
Point 19: Figures 7-8 should be both re-drawn with e.g., in Fig. 7 fewer digits after comma in the percentages and fig.8 improved the image’s quality
Response 19: Respected reviewers, thank you very much for your comments. We redraw Figure 7-8. Please review the following paragraphs,
Figure 7. Feature importance ranking of the XGBoost model; the sum of importance of all features is 100%. (Line 317-318)
Figure 8. Influence of the number of features on the model cross-validation mean. (Line 327-328)
Point 20: Line 310 – the use of past tense would be prefereable.
Response 20: Respected reviewers, thank you very much for your comments. We revised this sentence. Please review the following sentence,
As such, we found that the XGBoost model performs the best of the considered models. (Line 360-361)
Point 21: Line 314-317 this sentence does not have scientific relevance neither from an agronomic point of view nor from a modelling one.
Response 21: Respected reviewers, thank you very much for your comments. The SOM content in this study area is affected by the formula fertilization by soil testing policy. For details, please refer to the documents published by the Department of Agriculture and Rural Affairs of Anhui Province. We revised this sentence. Please review the following paragraph,
Anthrosols are the most widespread in the study area. The soil fertility is greatly affected by humans. The study area began to implement formula fertilization by soil testing policy in 2006. Government organized researchers to fertilize the cultivated land in the study area by measuring the physical and chemical composition of the soil (http://nync.ah.gov.cn/(accessed on December 8, 2021)). Due to the high proportion of anthrosols cultivated land in the study area(54.86% of the area), regular water and fertilizer adjustments make anthrosols more fertile than other soil types, and the spatial variability of the SOM content is low [76,77]. (Line 365-373)
Reference:
- Guo, N.; Shi, X.; Zhao, Y.; Xu, S.; Wang, M.; Zhang, G.; Wu, J.; Huang, B.; Kong, C. Environmental and anthropogenic factors driving changes in paddy soil organic matter: a case study in the Middle and Lower Yangtze River Plain of China. Pedosphere 2017, 27, 926-937.
- Duan, L.; Li, Z.; Xie, H.; Li, Z.; Zhang, L.; Zhou, Q. Large-scale spatial variability of eight soil chemical properties within paddy fields. CATENA 2020, 188, 104350, doi:https://doi.org/10.1016/j.catena.2019.104350.
Point 22: Line 320 a sentence with …”similar results in previous studies” does not discuss at all your results. Please contextualize and discuss more in depth your results with the ones found in other studies.
Response 22: Respected reviewers, thank you very much for your comments. We compared the results with those of other researchers. We revised this sentence. Please review the following paragraph,
In a similar study, researchers found that different land use types have different prediction accuracy of SOM content. By comparing the prediction accuracy of dryland and paddy fields, they proposed that the higher coefficient of variation of the dryland data set and the coefficient of variation of the paddy field data set may lead to better prediction accuracy of the dryland data set [78]. In this study, the coefficient of variation for the organic matter content of the anthrosols dataset was greater than that of the total data set. An increase in the coefficient of variation may improves the prediction accuracy of the model. (Line 373-381)
Reference:
- Wang, H.; Zhang, X.; Wu, W.; Liu, H. Prediction of Soil Organic Carbon under Different Land Use Types Using Sentinel-1/-2 Data in a Small Watershed. Remote Sensing 2021, 13, doi:10.3390/rs13071229.
Point 23: Line 327-328: this is a too general sentence that cannot be used to discuss the fact that vegetational indices had an influence on 2019 predicted results.
Response 23: Respected reviewers, thank you very much for your comments. We revised this sentence. Please review the following sentences,
The two largest vegetation indices in 2019 significantly influenced the prediction results of the model. The maximum vegetation index synthesized by GEE may be useful in future soil attribute prediction studies. (Line 387-391)
Point 24: Line 328-330: what are planting food crops ? Throughout all manuscript the terminology used for describing cropping systems and relative land use are inconsistent .
Response 24: Respected reviewers, thank you very much for your comments. We revised this sentence. Please review the following sentences,
The cultivated land planting situations were divided into planting food crops and the rotation of food and non-food crops. (Line 391-393)
Point 25: Line 331- as above “human adjustment of SOM” is inappropriate. SOM can be managed by farmers with organic fertilization, conservation tillage , targeted crop rotation etc… which are the other factor ? Please be more precise and consistent in result’s discussion.
Response 25: Respected reviewers, thank you very much for your comments. The cultivated land where planting crops in this study area is affected by the Organic Fertilizer Replacement policy. For details, please refer to the documents published by the Department of Agriculture and Rural Affairs of Anhui Province. Please review the following sentences,
The average SOM content of cultivated land for food crops is higher than that of cultivated land for rotation, and there are fewer abnormal values. This may be due to the organic fertilizer replacement policy implemented in the study area in 2016(http://nync.ah.gov.cn/(accessed on December 8, 2021)). (Line 393-397)
Point 26: Line 341 – “a lot of noise” please be more precise
Response 26: Respected reviewers, thank you very much for your comments. We revised this sentence. Please review the following sentences,
Multispectral satellite data are more suitable for the inversion of SOM content than hyperspectral satellite data. Due to the large number of reflection bands of hyperspectral data, it is necessary to eliminate redundant and cluttered bands with low correlation before establishing a regression model [50]. (Line 407-411)
Reference:
- Hong, Y.; Liu, Y.; Chen, Y.; Liu, Y.; Yu, L.; Liu, Y.; Cheng, H. Application of fractional-order derivative in the quantitative estimation of soil organic matter content through visible and near-infrared spectroscopy. Geoderma 2019, 337, 758-769, doi:https://doi.org/10.1016/j.geoderma.2018.10.025.
Point 27: Line 343 – what is “inversion research” ?
Response 27: Respected reviewers, thank you very much for your comments. We revised this sentence. Please review the following sentences,
GF-6 was the first high-resolution satellite developed by China for precision agriculture. Using GF-6 to predict the distribution of SOM content provides a basis for the development of precision agriculture policies. The province where the study area is located has been actively carrying out the construction of high-standard farmland since 2015, and clarifying the SOM content of the cultivated land is of great significance to the construction of high-standard farmland(http://nync.ah.gov.cn/(accessed on December 8, 2021)). (Line412-420)
Point 28: Line 345 – not clear. Please rephrase. What do “high-standard farmland” and “follow-up precise management” mean ?
Response 28: Respected reviewers, thank you very much for your comments. The construction of high-standard farmland policy is a major agricultural policy in this study area, the purpose is to increase food production. We revised this sentence. Please review the following sentences,
The province where the study area is located has been actively carrying out the construction of high-standard farmland since 2015, and clarifying the SOM content of the cultivated land is of great significance to the construction of high-standard farmland(http://nync.ah.gov.cn/(accessed on December 8, 2021)). (Line412-420)
We tried to improve the manuscript and made some changes in the manuscript. We appreciate for your warm work earnestly, and hope that the correction will meet with approval.
Once again, thank you very much for your comments and suggestions.
Reviewer 3 Report
Title: Using machine learning algorithms based on GF-6 and Google Earth Engine to predict and map the spatial distribution of soil organic matter content
The manuscript presented a case on building a remote sensing-based method for monitoring SOM content. Estimation of SOM is very important for measuring soil quality and the magnitude of degradation. This is a pertinent problem and worth investigation as conventional estimation approaches are inaccurate and time-consuming. Machine learning techniques such as XGBoost, LightGBM, GBDT and Random Forest were used for the model building process. Multispectral GF-6 satellite images, terrain, soil type and ground truth data were used as the model. The manuscript is nicely structured and well-documented. Hence, it may be recommended for acceptance after incorporating some minor revisions.
- Page: 2, line: 75, claiming something like “Support vector machines and random forest (RF) algorithms have been widely used in previous single-element inversions of SOM content.” needs to support with proper citation.
- What was the motivation behind the selection of modelling techniques for the analysis purpose? Why only decision tree-based algorithms were considered? Why were no techniques of different types such as deep learning techniques were included in the list?
- Grid search techniques have been used for tuning various model parameters. Although, a table with optimal parameters is included nothing is mentioned about different grid values that were considered for the search.
- How is this feature importance value generated in the case of the paddy soil dataset? I believe, XGBoost model was queried for this purpose as it performed the best. However, it is not clear that how this XGBoost model computes the importance level.
- How was runtime for different models were calculated? Is this a measure of time a model took to complete the training? If so, then runtime depends on the configuration of the system on which it was trained. Discussion in this regard is needed.
- Page: 16, line: 350, “Such models have low accuracy and 350 slow computational speed”. Is any reference available?
- The manuscript needs major revision in terms of sentence structuring. As of now, there are several instances where authors have written long sentences which are very difficult to comprehend. Some cases are:
- Page: 1, Line: 15: Usage of lengthy sentences needs to be avoided. For instance, “Based on the remote sensing data of China ……… to evaluate the prediction model”. Restructuring the sentence will help in improving the readability.
- Page: 6, line: 130: Meaning of the sentence “On this basis, the GEE synthesis algorithm was used to obtain the annual 130 maximum synthetic vegetation indices [55] in the study area pixel by pixel: Normalized Difference Vegetation Index (NDVI), Ratio Vegetation Index (RVI), Difference Vegetation Index (DVI), and 2018 Normalized Difference Water Index (NDWI)” is not clear. Please format it again.
There are many other instances available throughout the manuscript where lengthy sentences can be broken down to improve their quality.
Author Response
Response to Reviewer 3 Comments
Point 1: Page: 2, line: 75, claiming something like “Support vector machines and random forest (RF) algorithms have been widely used in previous single-element inversions of SOM content.” needs to support with proper citation.
Response 1: Respected reviewers, thank you very much for your comments. we added citation to this sentence. Please review the following sentence,
Support vector machines and random forest (RF) algorithms have been widely used in previous single-element SOM content inversions [22,46-50]. (Line 86-88)
Reference:
- Zhang, M.; Zhang, M.; Yang, H.; Jin, Y.; Zhang, X.; Liu, H. Mapping Regional Soil Organic Matter Based on Sentinel-2A and MODIS Imagery Using Machine Learning Algorithms and Google Earth Engine. REMOTE SENSING 2021, 13, doi:10.3390/rs13152934.
- Dong, Z.; Wang, N.; Liu, J.; Xie, J.; Han, J. Combination of machine learning and VIRS for predicting soil organic matter. JOURNAL OF SOILS AND SEDIMENTS 2021, 21, 2578-2588, doi:10.1007/s11368-021-02977-0.
- Wang, X.; Han, J.; Wang, X.; Yao, H.; Zhang, L. Estimating Soil Organic Matter Content Using Sentinel-2 Imagery by Machine Learning in Shanghai. IEEE ACCESS 2021, 9, 78215-78225, doi:10.1109/ACCESS.2021.3080689.
- Wang, Z.; Du, Z.; Li, X.; Bao, Z.; Zhao, N.; Yue, T. Incorporation of high accuracy surface modeling into machine learning to improve soil organic matter mapping. ECOLOGICAL INDICATORS 2021, 129, doi:10.1016/j.ecolind.2021.107975.
- Yang, J.; Li, X.; Wu, B.; Wu, J.; Sun, B.; Yan, C.; Gao, Z. High Spatial Resolution Topsoil Organic Matter Content Mapping Across Desertified Land in Northern China. FRONTIERS IN ENVIRONMENTAL SCIENCE 2021, 9, doi:10.3389/fenvs.2021.668912.
- Hong, Y.; Liu, Y.; Chen, Y.; Liu, Y.; Yu, L.; Liu, Y.; Cheng, H. Application of fractional-order derivative in the quantitative estimation of soil organic matter content through visible and near-infrared spectroscopy. Geoderma 2019, 337, 758-769, doi:https://doi.org/10.1016/j.geoderma.2018.10.025.
Point 2: What was the motivation behind the selection of modelling techniques for the analysis purpose? Why only decision tree-based algorithms were considered? Why were no techniques of different types such as deep learning techniques were included in the list?
Response 2: Respected reviewers, thank you very much for your comments. We added the motivation for choosing the model in the discussion, Please review the following paragraphs,
4.6. Model selection
Our research used the tree-based algorithm of machine learning. Tree-based algorithm models perform well in predicting linear problems, and researchers can easily interpret the model's predictions from the perspective of input features[44-47]. Deep learning models perform very well in the research of large data sets[82,83], but it is very difficult for researchers to explain the performance of the models in terms of their input features and model parameters[84]. In future research, we will try to use deep learning models to solve the problem of predicting the spatial distribution of SOM content. (Line 436-443)
Reference:
- Emadi, M.; Taghizadeh-Mehrjardi, R.; Cherati, A.; Danesh, M.; Mosavi, A.; Scholten, T. Predicting and Mapping of Soil Organic Carbon Using Machine Learning Algorithms in Northern Iran. REMOTE SENSING 2020, 12, doi:10.3390/rs12142234.
- Kobayashi, Y.; Yoshida, K. Quantitative structure?property relationships for the calculation of the soil adsorption coefficient using machine learning algorithms with calculated chemical properties from open-source software. ENVIRONMENTAL RESEARCH 2021, 196, doi:10.1016/j.envres.2020.110363.
- Dong, Z.; Wang, N.; Liu, J.; Xie, J.; Han, J. Combination of machine learning and VIRS for predicting soil organic matter. JOURNAL OF SOILS AND SEDIMENTS 2021, 21, 2578-2588, doi:10.1007/s11368-021-02977-0.
- Wang, X.; Han, J.; Wang, X.; Yao, H.; Zhang, L. Estimating Soil Organic Matter Content Using Sentinel-2 Imagery by Machine Learning in Shanghai. IEEE ACCESS 2021, 9, 78215-78225, doi:10.1109/ACCESS.2021.3080689.
- Szegedy, C.; Liu, W.; Jia, Y.; Sermanet, P.; Reed, S.; Anguelov, D.; Erhan, D.; Vanhoucke, V.; Rabinovich, A. Going deeper with convolutions. In Proceedings of Proceedings of the IEEE conference on computer vision and pattern recognition; pp. 1-9.
- Ciresan, D.; Giusti, A.; Gambardella, L.; Schmidhuber, J.J.A.i.n.i.p.s. Deep neural networks segment neuronal membranes in electron microscopy images. 2012, 25, 2843-2851.
- Montavon, G.; Lapuschkin, S.; Binder, A.; Samek, W.; Müller, K.-R. Explaining nonlinear classification decisions with deep Taylor decomposition. Pattern Recognition 2017, 65, 211-222, doi:https://doi.org/10.1016/j.patcog.2016.11.008
Point 3: Grid search techniques have been used for tuning various model parameters. Although, a table with optimal parameters is included nothing is mentioned about different grid values that were considered for the search.
Response 3: Respected reviewers, thank you very much for your comments. The specific value of the hyperparameter optimization search has been added to the table. Please review the following paragraphs,
Table 4. The optimal parameter, the default parameters and the range of the grid search in each model. (Line 275)
|
Regression Model |
Hyperparameters |
Optimal Parameter |
Default Parameters |
Range of Grid Search |
||
|
Extreme gradient boosting machine (XGBoost) |
random_state |
0 |
0 |
0 |
||
|
n_estimators |
15 |
100 |
0~100 |
|||
|
max_depth |
2 |
6 |
1~10 |
|||
|
learning_rate |
0.38 |
0.3 |
0.01~1.00 |
|||
|
min_child_weight |
4 |
1 |
1~10 |
|||
|
gamma |
0.1 |
0 |
0~1.0 |
|||
|
Light gradient boosting machine (LightGBM) |
random_state |
0 |
None |
0 |
||
|
n_estimators |
26 |
100 |
0~100 |
|||
|
max_depth |
7 |
-1 |
1~10 |
|||
|
learning_rate |
0.1 |
0.1 |
0.01~1.00 |
|||
|
subsample |
0.1 |
1 |
0~1.0 |
|||
|
Gradient boosting tree (GBDT) |
random_state |
0 |
None |
0 |
||
|
n_estimators |
21 |
100 |
0~100 |
|||
|
max_depth |
4 |
3 |
1~10 |
|||
|
learning_rate |
0.18 |
0.1 |
0.01~1.00 |
|||
|
Random Forest (RF) |
random_state |
0 |
None |
0 |
||
|
n_estimators |
83 |
100 |
0~100 |
|||
|
max_depth |
8 |
None |
1~10 |
|||
|
min_samples_split |
9 |
2 |
1~10 |
|||
|
min_samples_leaf |
1 |
1 |
1~10 |
|||
Point 4: How is this feature importance value generated in the case of the paddy soil dataset? I believe, XGBoost model was queried for this purpose as it performed the best. However, it is not clear that how this XGBoost model computes the importance level.
Response 4: Respected reviewers, thank you very much for your comments. We have supplemented how the value of feature importance is derived. Please review the following paragraphs,
3.4.2. Feature selection for anthrosols dataset
Feature importance is an important reference when selecting features. The XGBoost model uses the gain criterion to calculate the importance of each feature when participating in model training. The gain is calculated by the contribution of the feature to each tree, that is, the contribution of each feature to the generative model. The higher the value, the greater the importance of this feature to the prediction of the model [75]. (Line 308-316)
Reference:
- Zhang, W.; Wu, C.; Zhong, H.; Li, Y.; Wang, L. Prediction of undrained shear strength using extreme gradient boosting and random forest based on Bayesian optimization. Geoscience Frontiers 2021, 12, 469-477, doi:10.1016/j.gsf.2020.03.007.
Point 5: How was runtime for different models were calculated? Is this a measure of time a model took to complete the training? If so, then runtime depends on the configuration of the system on which it was trained. Discussion in this regard is needed.
Response 5: Respected reviewers, thank you very much for your comments. We have supplemented the calculation speed of the model. Please review the following paragraphs,
3.3.2. Evaluation of model calculation speed
Our experimental computer is based on Windows 10 system, Core i7 10710 processor, 16G RAM. Different machine-learning models exhibit different computing speeds. We compared the calculation speed of different models by comparing the calculation completion time of different models on the same computer platform. We found that RF model was the slowest of the four machine learning models, and the LightGBM model was the fastest computing model. (Line 278-284)
Point 6: Page: 16, line: 350, “Such models have low accuracy and 350 slow computational speed”. Is any reference available?
Response 6: Respected reviewers, thank you very much for your comments. We added citation to this sentence. Please review the following sentence,
Such models have low accuracy and slow computational speed [38,50]. (Line 424-425)
Reference:
- Fu, C.; Gan, S.; Yuan, X.; Xiong, H.; Tian, A. Impact of Fractional Calculus on Correlation Coefficient between Available Potassium and Spectrum Data in Ground Hyperspectral and Landsat 8 Image. Mathematics 2019, 7, doi:10.3390/math7060488.
Point 7: The manuscript needs major revision in terms of sentence structuring. As of now, there are several instances where authors have written long sentences which are very difficult to comprehend. Some cases are:
Page: 1, Line: 15: Usage of lengthy sentences needs to be avoided. For instance, “Based on the remote sensing data of China ……… to evaluate the prediction model”. Restructuring the sentence will help in improving the readability.
Page: 6, line: 130: Meaning of the sentence “On this basis, the GEE synthesis algorithm was used to obtain the annual 130 maximum synthetic vegetation indices [55] in the study area pixel by pixel: Normalized Difference Vegetation Index (NDVI), Ratio Vegetation Index (RVI), Difference Vegetation Index (DVI), and 2018 Normalized Difference Water Index (NDWI)” is not clear. Please format it again.
Response 7: Respected reviewers, thank you very much for your comments. We have broken down the long sentences of the article. Please review the following paragraphs,
The prediction of soil organic matter is important for measuring the soil’s environmental quality and the degree of degradation. In this study, we combined China’s GF-6 remote sensing data with the organic matter content data obtained from soil sampling points in the study area to predict soil organic matter content. To these data, we applied the random forest (RF), light gradient boosting machine (LightGBM), gradient boosting tree (GBDT), and extreme boosting machine (XGBoost) learning models. We used the coefficient of determination (R2), root mean square error (RMSE), and mean absolute error (MAE) to evaluate the prediction model. (Line 15-22)
On this basis, we used the GEE synthesis algorithm to obtain the following maximum synthetic vegetation indices [58] annually pixel by pixel in the study area: the normalized difference vegetation index(NDVI), ratio vegetation index(RVI), difference vegetation index(DVI), and 2018 normalized difference water index(NDWI). As the input feature of the model, we extracted the maximum synthetic vegetation index of the ground sample points. (Line 149-156)
In this study, we separately trained four machine learning models. From the observed performance of the model and the results of the training time, most hyperparameters need to be adjusted for the RF model, and the model’s prediction accuracy and training time performance are the worst. LightGBM has the fastest training time, but it has certain dataset, and its stability is low. XGBoost improves the GBDT model at the algorithm level, and its prediction accuracy and efficiency are higher than those of the GBDT model. As such, we found that the XGBoost model performs the best of the considered models. (Line 353-362)
We tried to improve the manuscript and made some changes in the manuscript. We appreciate for your warm work earnestly, and hope that the correction will meet with approval.
Once again, thank you very much for your comments and suggestions.

Round 2
Reviewer 2 Report
The authors successfully addressed most of the reviewer's comments and answered the main questions. However, points 24, 25 and 28 have been not answered. The reason on the use of "food crops", the discussion on the SOM contents between land uses and the future application of such estimation model are not satisfactory yet. Furthermore, a description of the meaning of "high standard farmland" was not provided.
Author Response
Response to Reviewer 2 Comments
Point 24: Line 328-330: what are planting food crops? Throughout all manuscript the terminology used for describing cropping systems and relative land use are inconsistent.
Response 24: Respected reviewers, thank you very much for your comments. We didn't have a clear definition of cultivated land and food crops before. Food crops in China refer to rice, wheat, and corn. Cultivated land refers to dry land, paddy field and irrigated land. According to "Classification of Land Use Status" (GB/T21010-2007): Cultivated land refers to the land on which crops are grown, including cultivated land, newly developed, reclaimed, and reorganized land, and fallow land (including rotation land). We revised these sentences. Please review the following sentences,
Cultivated land refers to dry land, paddy field and irrigated land. The cultivated land planting situation of the sample points in the study area is largely divided into planting food crops and rotation with non-food crops. Food crops in the study area refer to rice, wheat, and corn. Non-food crops in the study area refer to vegetables and fruits. (Line 166-170)
The cultivated land planting situations were divided into planting food crops and the rotation of food and non-food crops. (Line 391-393)
Point 25: Line 331- as above “human adjustment of SOM” is inappropriate. SOM can be managed by farmers with organic fertilization, conservation tillage , targeted crop rotation etc… which are the other factor ? Please be more precise and consistent in result’s discussion.
Response 25: Respected reviewers, thank you very much for your comments. The cultivated land where planting crops in this study area is affected by the Organic Fertilizer Replacement policy. For details, please refer to the documents published by the Department of Agriculture and Rural Affairs of Anhui Province. Please review the following sentences,
The average SOM content of cultivated land for food crops is higher than that of cultivated land for rotation, and there are fewer abnormal values. This may be due to the organic fertilizer replacement policy implemented in the study area in 2016(http://nync.ah.gov.cn/(accessed on December 8, 2021)). Farmers in the study area are required to use organic fertilizers instead of part of the chemical fertilizers applied to the cultivated land where food crops are grown. This policy started in 2016, replacing chemical fertilizers with organic fertilizers can increase the SOM content [79]. (Line 395-401)
Reference:
- Li, Z.Q.; Zhang, X.; Xu, J.; Cao, K.; Wang, J.H.; Xu, C.X.; Cao, W.D. Green manure incorporation with reductions in chemical fertilizer inputs improves rice yield and soil organic matter accumulation. Journal of Soils and Sediments 2020, 20, 2784-2793, doi:10.1007/s11368-020-02622-2.
Point 28: Line 345 – not clear. Please rephrase. What do “high-standard farmland” and “follow-up precise management” mean ?
Response 28: Respected reviewers, thank you very much for your comments. We made a mistake in translation. High-standard farmland should be translated as well-facilitated farmland. The construction of well-facilitated farmland policy is a major agricultural policy in this study area. We revised this sentence. Please review the following sentences,
The province where the study area is located has been actively carrying out the construction of well-facilitated farmland [83] since 2015, and clarifying the SOM content of the cultivated land is of great significance to the construction of well-facilitated farmland (http://nync.ah.gov.cn/(accessed on December 8, 2021)). Well-facilitated farmland refers to farmland with leveled land, concentrated contiguous areas, complete facilities, supporting farmland, fertile soil, good ecology, strong disaster resistance, and is compatible with modern agricultural production and management methods that are suitable for droughts and floods, and high and stable yield farmland. The results of this study can be used to monitor the SOM content of farmland when constructing and evaluating well-facilitated farmland. (Line423-432)
Reference:
- Wang, X.; Shi, W.; Sun, X.; Wang, M. Comprehensive benefits evaluation and its spatial simulation for well-facilitated farmland projects in the Huang-Huai-Hai Region of China. Land Degradation & Development 2020, 31, 1837-1850, doi:10.1002/ldr.3566.
We tried to improve the manuscript and made some changes in the manuscript. We appreciate for your warm work earnestly, and hope that the correction will meet with approval.
Once again, thank you very much for your comments and suggestions.

Reviewer 3 Report
The revision made is up to the mark. Authors did a good job.
Author Response
Thank you very much for your comments. Your comments are very important for us to perfect the manuscript. Wish you happy everyday.
This manuscript is a resubmission of an earlier submission. The following is a list of the peer review reports and author responses from that submission.